# Dupilumab-associated head and neck dermatitis shows a pronounced type 22 immune signature mediated by oligoclonally expanded T cells

Christine Bangert [1,5], Natalia Alkon [1,5], Sumanth Chennareddy[2], Tamara Arnoldner[1], Jasmine P. Levine[2,3], Magdalena Pilz[1], Marco A. Medjimorec[1], John Ruggiero[2], Emry R. Cohenour[2], Constanze Jonak [1], William Damsky [4], Johannes Griss [1] & Patrick M. Brunner [2] ✉

Dupilumab, an IL4R-blocking antibody, has shown clinical efficacy for atopic dermatitis (AD) treatment. In addition to conjunctivitis/blepharitis, the de novo appearance of head/neck dermatitis is now recognized as a distinct side effect, occurring in up to 10% of patients. Histopathological features distinct from AD suggest a drug effect, but exact underlying mechanisms remain unknown. We profiled punch biopsies from dupilumab-associated head and neck dermatitis (DAHND) by using single-cell RNA sequencing and compared data with untreated AD and healthy control skin. We show that dupilumab treatment was accompanied by normalization of IL-4/IL-13 downstream activity markers such as *CCL13, CCL17, CCL18* and *CCL26*. By contrast, we found strong increases in type 22-associated markers (*IL22, AHR*) especially in oligoclonally expanded T cells, accompanied by enhanced keratinocyte activation and IL-22 receptor upregulation. Taken together, we demonstrate that dupilumab effectively dampens conventional type 2 inflammation in DAHND lesions, with concomitant hyperactivation of *IL22*-associated responses.

Atopic dermatitis (AD), the most common chronic inflammatory skin disease, has long been a therapeutic challenge, especially in its moderate-to-severe form[1]. Due to the recent advent of novel targeted therapeutics, treatment options for these patients have improved considerably[2–4]. The success of these agents not only contributed to the unraveling of many cellular and molecular mechanisms relevant to AD pathogenesis, but also resulted in an increased overall understanding of skin immunobiology[1,5–11]. Nevertheless, certain aspects of AD pathophysiology are still only incompletely understood. One surprising feature is the appearance of paradoxical pro-inflammatory reactions in a subset of AD patients treated with the IL-4Rα blocker dupilumab[12,13], which has not been observed in individuals with other type 2 diseases, such as allergic asthma[14]. These side effects include dupilumab-induced blepharitis and conjunctivitis, as well as dupilumab-associated head and neck dermatitis (DAHND)[15]. While eye involvement is now assumed to result from goblet cell scarcity and impaired IL-13-dependent mucus production[16,17], mechanisms behind the appearance of DAHND are still largely unclear. In contrast to the mere persistence of pre-existing AD ("residual facial dermatitis during dupilumab therapy")[18–20], which is a known phenomenon in difficult-to-treat AD patients[21,22], the de novo appearance of DAHND is now recognized as a distinct entity, which can be found in up to 10% of AD

[1]Department of Dermatology, Medical University of Vienna, Vienna, Austria. [2]Icahn School of Medicine at Mount Sinai, New York City, NY, USA. [3]New York Medical College, Valhalla, NY, USA. [4]Department of Dermatology, Yale School of Medicine, New Haven, CT, USA. [5]These authors contributed equally: Christine Bangert, Natalia Alkon. ✉e-mail: patrick.brunner@mountsinai.org

patients treated with dupilumab[12,23–25]. This notion is supported by histopathological differences from conventional AD[23,24], as well as the fact that DAHND has been described to resolve when patients are switched from dupilumab to the JAK1 inhibitor upadacitinib[26]. Responses to antifungal treatment in some case series supports Malassezia hypersensitivity as a triggering factor[26–29], but other suggested mechanisms include psoriasiform or rosacea-like reactions, as well as concomitant allergic contact dermatitis[25,30–32].

To better understand pathogenic mechanisms of DAHND, we characterize cellular and molecular signatures of full-thickness skin biopsies from DAHND and compare our results with untreated AD lesions both from the head/neck area and from the trunk, as well as to skin from healthy individuals, which serve as a baseline control. By doing so, we not only show that dupilumab treatment in DAHND lesions is accompanied by normalization of IL-4/IL-13 downstream activity markers such as *CCL13, CCL17, CCL18,* and *CCL26*, but also identify strong increases in type 22-associated inflammation especially in oligoclonally expanded T cells, accompanied by enhanced keratinocyte activation (*S100A7, S100A8, S100A9*) and IL-22 receptor upregulation.

## Results

### Single-cell RNA sequencing map of skin biopsies comparing DAHND with untreated AD and healthy control skin

By using single-cell RNA sequencing (scRNA-seq), we profiled a total of 196,040 cells from samples of AD patients suffering from DAHND (70,102, n = 6, Fig. 1A) and compared results with untreated AD of the head/neck region (59,066, n = 5), untreated AD of the trunk (51,038, n = 5), and skin from healthy control (HC) individuals (15,834, n = 4) (Supplementary Data 1). Clustering analyses identified 41 distinct cell populations (Fig. 1B, C, Supplementary Data 2–3), which were labeled according to the presence of canonical cell markers (Fig. 1D). Both untreated AD groups and DAHND samples showed generally increased numbers of leukocytes and keratinocytes when compared to HC (Fig. 1C), consistent with chronic inflammation and acanthosis, respectively. Proliferating cells (pro), as well as *CRTAM + CD8A +* T cells (T6), *LAMP3+* dendritic cells (DC4) and mast cells (MC) were essentially absent in HC (Fig. 1C–E, Supplementary Data 2). The initial "B" cluster could be further subdivided into plasma cells (PC), B cells (BC) and plasmacytoid dendritic cells (pDCs), with significant increases in *MS4A1+ CD79A+ CD27-* B cell and *MS4A1-CD79A+ CD27+* plasma cell counts in DAHND (Fig. 2A–F) that could be confirmed in multicolor immunofluorescence stainings in an independent sample set (Fig. 2G, H, Supplementary Data 1).

### DAHND lesions are characterized by enhanced type 22-associated inflammation

Among the lymphoid population, we found a large *CD4+* helper T-cell cluster with some admixed *CD8A+* cells (T1), as well as clusters primarily consisting of *CD4+* helper T cells (T2) and *FOXP3+ IL2RA+* regulatory T cells (T3) (Fig. 3A–C). We also found large *CD8A+* cytotoxic T cell populations T4 and T5, a smaller cytotoxic *CRTAM+ CCL1+* cluster T6, and small clusters of *MKI67+* proliferating T cells (Tpro), *NKG7+* NK cells and *CD3D-KLRB1+ KLRD1-* ILCs (Fig. 3A–C). T5 differed from other *CD8A+* T cells by the expression of chemokine ligands *CCL4* and *CCL5*, similar to NK cells (Fig. 3B, C, Supplementary Data 3). Helper T cells in T2 and regulatory T cells in T3 were essentially devoid of pro-inflammatory T cell lead cytokines, while T5 and NK cells expressed the type 1-associated marker *IFNG* (Fig. 3C). The prototypic type 2 cytokine *IL13* was most strongly expressed in the *CRTAM+ CD8A+* T6 cluster and the mixed T1 population, followed by *CD8A+* cytotoxic T cells in T4 and proliferating T cells (Fig. 3C). Other T cell subset-defining cytokines of the type 17[33] (*IL17A/IL17F*) or type 9 lineage[34] (*IL9*) were largely absent, whereas *IL13*-producing T cell clusters T1, T4, T6, and proliferating T cells were also positive for *IL22, IL26,* and *CSF2*, with strongest

expression in the small T6 cluster (Fig. 3C). T-cell subsets of special interest, such as *CRTAM+ CD8A+* T6 and *PTGDR2+ IL17RB+* "Th2A" cells, previously associated with the AD skin disease memory[35], as well as proliferating T cells, were absent in healthy control individuals, but present in all AD groups and in DAHND (Fig. 3D, E). In line, the inflammatory cytokines *IL13, IL22,* and *IL26* were largely absent in HC (Fig. 3F–I). Despite ongoing type 2 inhibition in dupilumab-treated samples, *IL13* was still present at substantial levels in DAHND-derived T cells (Fig. 3F–I). When calculating differential gene expression among all groups (Supplementary Data 4) or by pairwise comparison of DAHND with location-matched head/neck AD (Supplementary Data 5), DAHND samples demonstrated the strongest expression of *IL22* in several T cell subsets, particularly T1 and T4 (Fig. 3F, G, Supplementary Data 4-5). Consistent with type 22 polarization, these DAHND T cell clusters showed increases in expression of the aryl hydrocarbon receptor *AHR* (Fig. 3F–I, Supplementary Data 4–5), the key Th22/Tc22 transcription factor[36]. To corroborate these single-cell findings, we performed quantitative RT-PCR of key T cell cytokines (Figure S1) in an independent set of skin biopsies (Supplementary Data 1). In line, we found persisting *IL13* expression in DAHND with further elevation of *IL22* levels, but only low expression of *IL17A, IL17F, IFNG,* and *IL10* that was in the range of HC skin (Figure S2).

Despite this highly pro-inflammatory environment with enhanced type 22 immune responses, several inflammatory mediators were normalized in DAHND lesions towards HC levels. Expression of the tumor suppressor *PDCD4*, which was downregulated in untreated AD, was restored in DAHND lesions (Figure S2). *LIME1* (Lck-interacting transmembrane adapter 1), previously shown to be involved in chemokine-mediated migration of effector T cells to inflammatory sites[37], was also downregulated in several T cell clusters of DAHND (Figure S2). Despite sustained levels of *IL13* gene expression, we found *ITM2A*, a target gene of the prototypical type 2 transcription factor GATA3[38], as well as the type 2 associated marker *AREG*[39], to be significantly downregulated in multiple T cell subsets in DAHND lesions (Figure S2), confirming overall dampened type 2 immune responses. In contrast to the normalization or downregulation of these inflammatory mediators, we found increased expression of the keratinocyte adhesion molecule desmoyokin *AHNAK* in DAHND, an enhancer of overall T cell function (Figure S2)[40,41].

### DAHND lesions show strongest expansion of *IL22/IL13/AHR* expressing T cell oligoclones

To better understand clonotypic properties of DAHND lesions, we combined T cell receptor sequencing with our scRNA-seq dataset and quantified total numbers of the top 10 expanded T cell clones per sample (Fig. 4A). While top clone numbers were increased above healthy control samples in all AD groups, consistent with previous reports[42–44], the highest frequencies of oligoclonal expansion were found in DAHND lesions, outnumbering all other groups (Fig. 4A). Importantly, the top 10 expanded clones in HC samples were non-specifically distributed across T cell clusters at low numbers, whereas in all AD and DAHND groups, the top 10 expanded clones were particularly increased in T1 and T4 (Fig. 4B). When assessing genes differentially expressed between the 10 oligoclonally expanded T cells and the remaining polyclonal T cell infiltrate, we detected 82 mutually regulated genes across disease groups, i.e., 70 up- and 12 downregulated genes (Fig. 4C, Supplementary Data 6). While polyclonal cells were high in *LTB, CCL5* or the central T cell-memory marker *SELL*, upregulated genes of top oligoclones included *CD8A, CD8B, IL9R,* and *KLRK1* (killer cell lectin like receptor K1), the type 2 key cytokine *IL13*, but also *IL22* and *AHR* (Fig. 4C, Supplementary Data 6). In line, a pattern of primarily *CD8A/CD8B* expansion was visible among a line-up of the individual top 10 TCR clones of most untreated AD and DAHND patients (Fig. 4D). As a next step, we performed confirmatory immunofluorescence stainings of the combination of surface markers CD8, CD129 (*IL9R*) and CD314

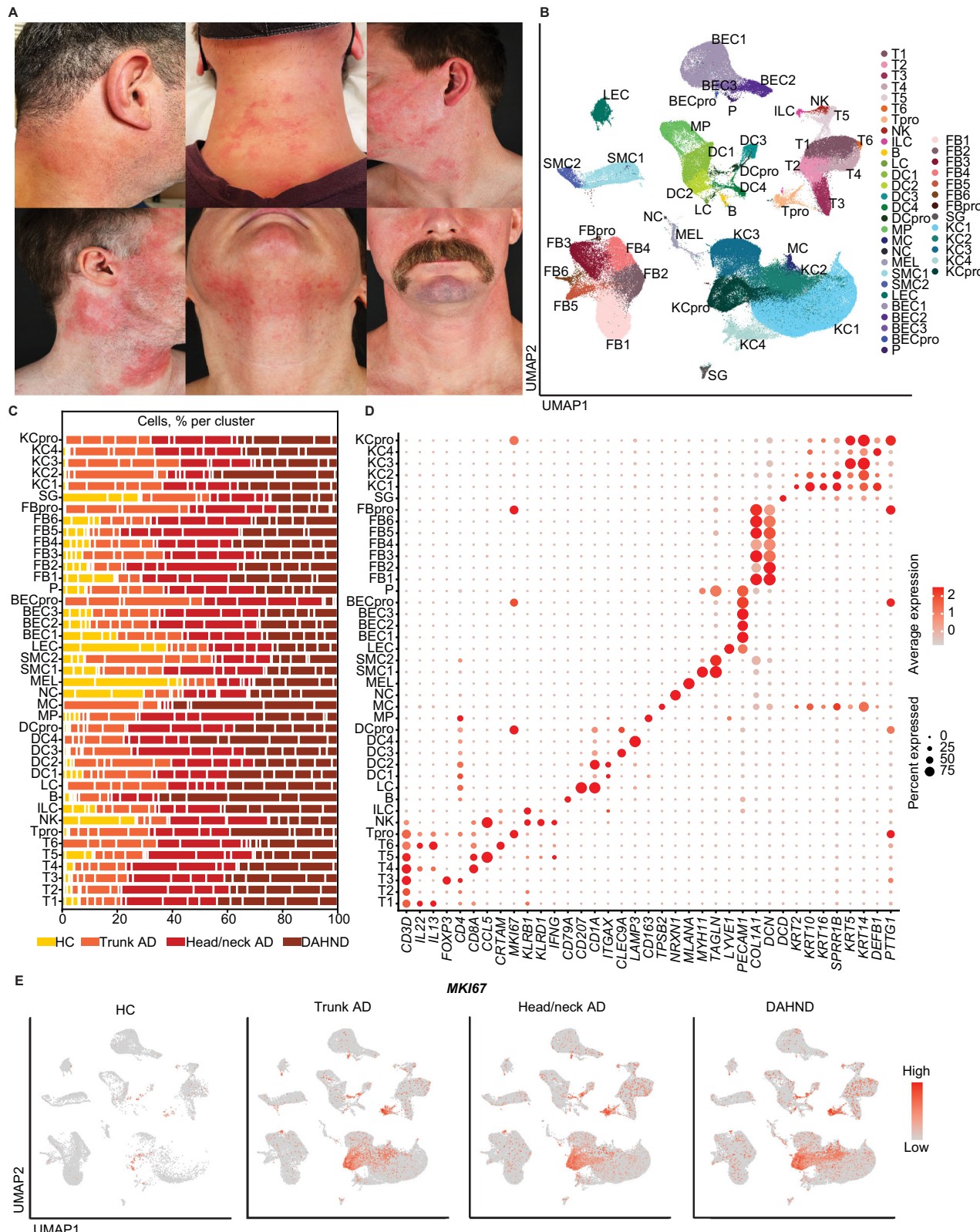

(*KLRK1*), characteristic for these top 10 oligoclones (Fig. 4C), by using an independent set of biopsy samples from HC, trunk AD, head/neck AD and DAHND (Supplementary Data 1). We found a significant increase of triple-positive cells in DAHND lesions (Fig. 4E, F), corroborating scRNA-seq data. Importantly, the distribution pattern of oligoclonal T cells within scRNA-seq clusters (Fig. 4G) was largely congruent with *IL13, IL22,*

*IL26,* and *CSF2* expression (Fig. 4H), with trunk AD oligoclones and cytokine expression being located more to the right side of the cluster, while head/neck and DAHND samples spanned the entire T1/T4 area (Fig. 4H). Remarkably, overall gene expression of *IL13, IL22,* and *AHR* were highest in DAHND oligoclones, while those from untreated head/ neck AD characteristically showed upregulation of the type 2-associated

**Fig. 1 | scRNA-seq map of cells from DAHND, untreated head/neck AD and trunk AD, as well as HC samples. A** Clinical pictures of the 6 patients that were included in this study with de novo dupilumab-associated head and neck dermatitis (DAHND). **B** UMAP plot of unsupervised clustering of 196,040 cells integrated from DAHND ($n = 6$), untreated head/neck AD ($n = 5$), untreated trunk AD ($n = 5$) and HC samples ($n = 4$) according to similarity of transcriptome, resulting in 41 different color-coded clusters. **C** Relative distribution of samples per condition across cell clusters. **D** Dot plot displaying average gene expression and frequency of canonical cell type markers for each cluster. Circle size represents the percentage of cells

expressing the specific marker within a cluster. Coloring denotes expression levels within each cluster (red is high). **E** Feature plots of disease groups showing the proliferation marker *MKI67*. Intensity of expression levels for each cell is color-coded (red) and overlaid onto UMAP plots. UMAP Uniform Manifold Approximation and Projection, T T-cells, NK NK cells, ILC innate lymphoid cells, B cells of the B cell lineage, LC Langerhans cells, DC dendritic cells, MP macrophages, MC mast cells, NC neuron cells, MEL melanocytes, SMC smooth muscle cells, LEC lymphatic endothelial cells, BEC blood endothelial cells, P pericytes, FB fibroblasts, SG sweat gland cells, KC keratinocytes; "pro" indicates proliferating subsets.

marker amphiregulin *AREG*[39], and those from untreated trunk AD showed highest levels of *IL26, CCL1* and *CSF2* (Fig. 4I, Supplementary Data 7). Consistent with scRNA-seq, we found elevated numbers of CD8+CD129+IL22+ cells in immunofluorescence stainings of DAHND lesions within our independent sample set (Fig. 4J, K). Taken together, these data confirm an overall highly inflammatory milieu in DAHND lesions that is likely triggered by specifically expanded type 22 CD8+ T cells.

### Dendritic cells show decreased type 2 responses in DAHND consistent with functionally relevant IL-4Rα blockade within the tissue microenvironment

Besides lymphoid cells, the largest immune cell cluster comprised several groups of myeloid cells. These included *CD68+ ITGAX-* macrophages (MP), *CD1C+ CD1A-* dendritic cells DC1, *CD1A+ CD1B+ CD1C+ FCER1A+ MRC1+* dendritic cells DC2 which were most consistent with inflammatory dendritic epidermal cells (IDECs)[45,46], *CLEC9A+ XCR1+* cross-presenting DC3, *LAMP3+* mature DC4, *MKI67+* proliferating DCs (DCpro) and *CD207+* Langerhans cells (LC) (Fig. 5A, B).

Generally, myeloid cell numbers were increased in AD and DAHND lesions compared to HC samples (Fig. 5C). Macrophages were most reminiscent of an M2 phenotype, due to upregulation of *MRC1, CD163, MS4A4A, CD209*, and lack of *CD80, CD86, FCGR1A, CD38, GPR18*, and *FPR2* expression (Fig. 5B). The type 2 associated chemokines *CCL13, CCL17, CCL18*, and *CCL22* were largely confined to MP, DC2, and/or DC4 (Fig. 5B). While being highly expressed in untreated AD samples of the trunk and head/neck area, these chemokines were consistently downregulated in DAHND (Fig. 5D, Supplementary Data 4), suggesting active inhibition of type 2-associated inflammation by dupilumab, despite retained *IL13* upregulation in multiple T cell populations (Fig. 3). DAHND DCs also showed downregulation of chemokine-like factor *CKLF* (Fig. 5E), a mediator known to be overexpressed in AD and previously assumed to play a role in the recruitment of type 2 cells via CCR4 ligation[47].

Markers uniquely regulated in dendritic cells of trunk AD included decreases in the pro-inflammatory innate mediator *IL1B* and upregulation of the matrix metalloproteinase *MMP12* and the lysosomal acid lipase *LIPA* (Fig. 5F, G), suggesting a distinct immune microenvironment depending on body location. In line, *COTL1*, a stabilizer of the 5-lipoxygenase *ALOX5*, showed lower levels in trunk AD compared to samples from the head/neck region (Fig. 5H), which may facilitate a generally stronger inflammatory milieu in the head/neck region compared to the trunk.

Consistent with overall enhanced inflammation in DAHND, several DC clusters showed downregulation of the lipoxygenase *ALOX15*, a molecule implicated in anti-inflammatory properties of the skin[48], in comparison to untreated AD groups (Fig. 5I). DAHND DC4 also showed increased expression of *RARRES2* (Fig. 5J, Supplementary Data 4), a gene encoding the pro-inflammatory protein chemerin that is typically upregulated in early psoriatic lesions[49], which has also been shown to be a strong chemotactic factor for dendritic cells[50]. The serine proteinase inhibitor *SERPINB9*, previously implicated as a cytoprotective mediator by opposing the cytotoxic effects of granzyme B[51], was upregulated in DAHND DC2 and DC4 clusters (Fig. 5K), thereby potentially harnessing these cells against cytotoxic T cell effects.

In parallel with enhanced *IL22* activity by T cells (Figs. 3 and 4), LC showed significant downregulation of the IL-22 scavenger *IL22RA2* in DAHND lesions (Fig. 5L), a gene coding for IL-22 binding protein (IL-22BP). Given the fact that IL-22BP has previously been described as a major inhibitor of IL22 function[52], we performed multicolor immunofluorescence stainings in our independent sample set (Supplementary Data 1). We could corroborate that a small subset of CD207+ LC co-expressed IL-22BP on the protein level in all conditions (Fig. 6A–D). However, we observed a dramatic increase of CD207+ LCs in the dermis of DAHND samples that was not present in untreated AD groups or HC (Fig. 6E). Importantly, only a minor proportion of these dermal CD207+ cells in DAHND co-expressed IL-22BP (Fig. 6F). Overall, these data confirm effective suppression of type 2 signaling in DAHND despite ongoing *IL13* production, with increases in dermal LC populations that are likely no longer able to counteract *IL22*-associated skin inflammation.

### Keratinocytes show IL-22-associated hyperactivation

Among keratinocytes, clusters spanned *MKI67+* proliferating cells (KCpro), *COL7A1+ KRT5+ KRT14+* basal KC3, as well as suprabasal *KRT1+ KRT10+* populations KC1 and KC2 (Fig. 7A, B). We also found a cluster KC4 that expressed *CHI3L1*, previously described as a driver of allergic skin inflammation[53], and *MGST1*, a marker of sebocytes[54]. In addition, we detected a smaller population of sweat gland cells (SG) that characteristically showed expression of *DCD* and *KRT18* (Fig. 7B, Supplementary Data 3). There were generally elevated keratinocyte counts in disease groups in comparison to HC samples (Fig. 7C). In line with a type 22-biased inflammatory milieu, keratinocytes of DAHND showed strongest expression of the IL-22-induced[55] keratinocyte hyperactivation markers *S100A7, S100A8*, and *S100A9*, that were absent in HC samples (Fig. 7D, Supplementary Data 4). Importantly, a pseudo-bulk analysis of the entire dataset confirmed that in DAHND compared to all other groups, the most abundantly upregulated genes were *S100A7* and *S100A9*, followed by *IL22* (Supplementary Data 8). Consistent with dupilumab-associated inhibition of type 2 cytokines[7], we found downregulation of inflammatory keratins *KRT16, KRT6A, KRT6B*, and *KRT6C*, and upregulation of the barrier markers *IVL* and *SPINK5* in keratinocytes of DAHND compared to untreated AD groups (Fig. 7D). Similarly, the keratinocyte differentiation markers *CALML3* and *CALML5*[56] were upregulated in DAHND (Fig. 7D). *POSTN*, known to be induced by type 2 inflammation[57], was downregulated in DAHND lesions (Fig. 7D). We also detected decreased expression of *CCL27* in DAHND keratinocytes, a chemokine ligand classically downregulated in psoriatic inflammation and upregulated in AD[58]. Additionally, genes from a previously published type 2-specific keratinocyte response signature[59] corroborated downregulation of the type 2 signature in DAHND keratinocytes compared to untreated AD (Fig. 7E). Taken together, these data confirm substantial reduction in type 2 inflammation in DAHND.

In parallel to reduced type-2 associated inflammatory patterns, we also found downregulation of the gamma interferon-induced markers *IFITM2* and *IFITM3*[60] in DAHND, with the highest levels in untreated trunk AD (Fig. 7D). In line with enhanced IL-22-dependent inflammation in DAHND, we found *SOCS3*, previously implicated in inhibition of IL-22 effects on NMSC[61], to be present at significantly lower levels in DAHND compared to untreated head/neck AD (Fig. 7D). Also, *LGALS7*, a selectin

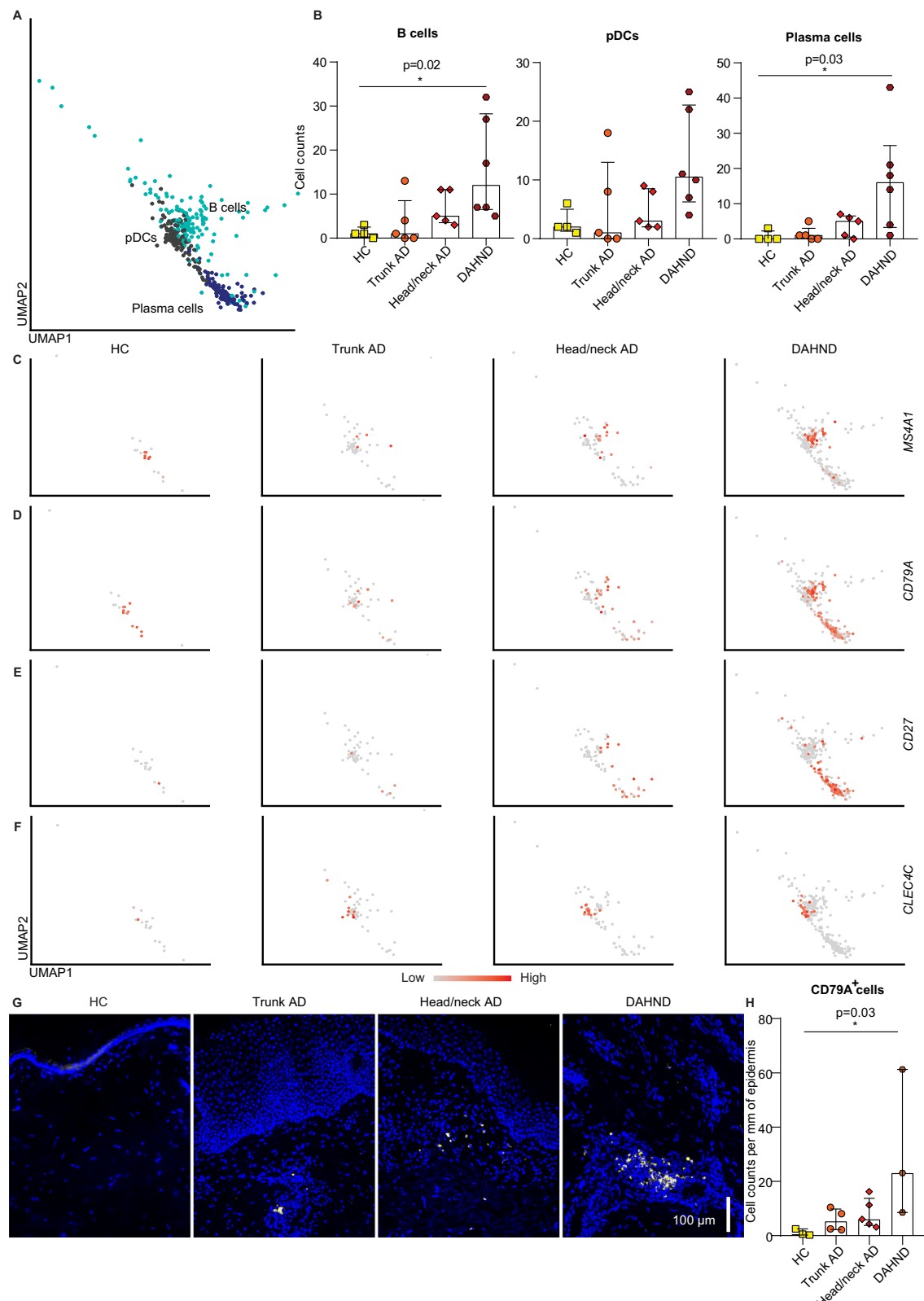

**Fig. 2 | Subclustering of the larger "B cell" cluster. A** Subclustering into B cells, plasmacytoid dendritic cells (pDCs), and plasma cells. **B** Absolute cell counts across conditions: HC (*n* = 4), trunk AD (*n* = 5), head/neck AD (*n* = 5), and DAHND (*n* = 6); each dot represents a single individual; data are presented as median with interquartile range (IQR). **C**–**F** Feature plots of selected gene expression of canonical markers. Intensity of expression levels for each cell is color-coded (red is high) and overlaid onto UMAP plots. **G**, **H** Representative immunofluorescence stainings of the B cell marker CD79A as well as absolute cell counts of CD79A+ cells per mm epidermis across conditions: HC (*n* = 3), trunk AD (*n* = 4), head/neck AD (*n* = 5), and DAHND (*n* = 3); each dot represents a single individual; data are presented as median with IQR. Statistical significance in this figure was calculated using a Kruskal–Wallis test for multiple comparisons followed by a Dunn's post hoc test.

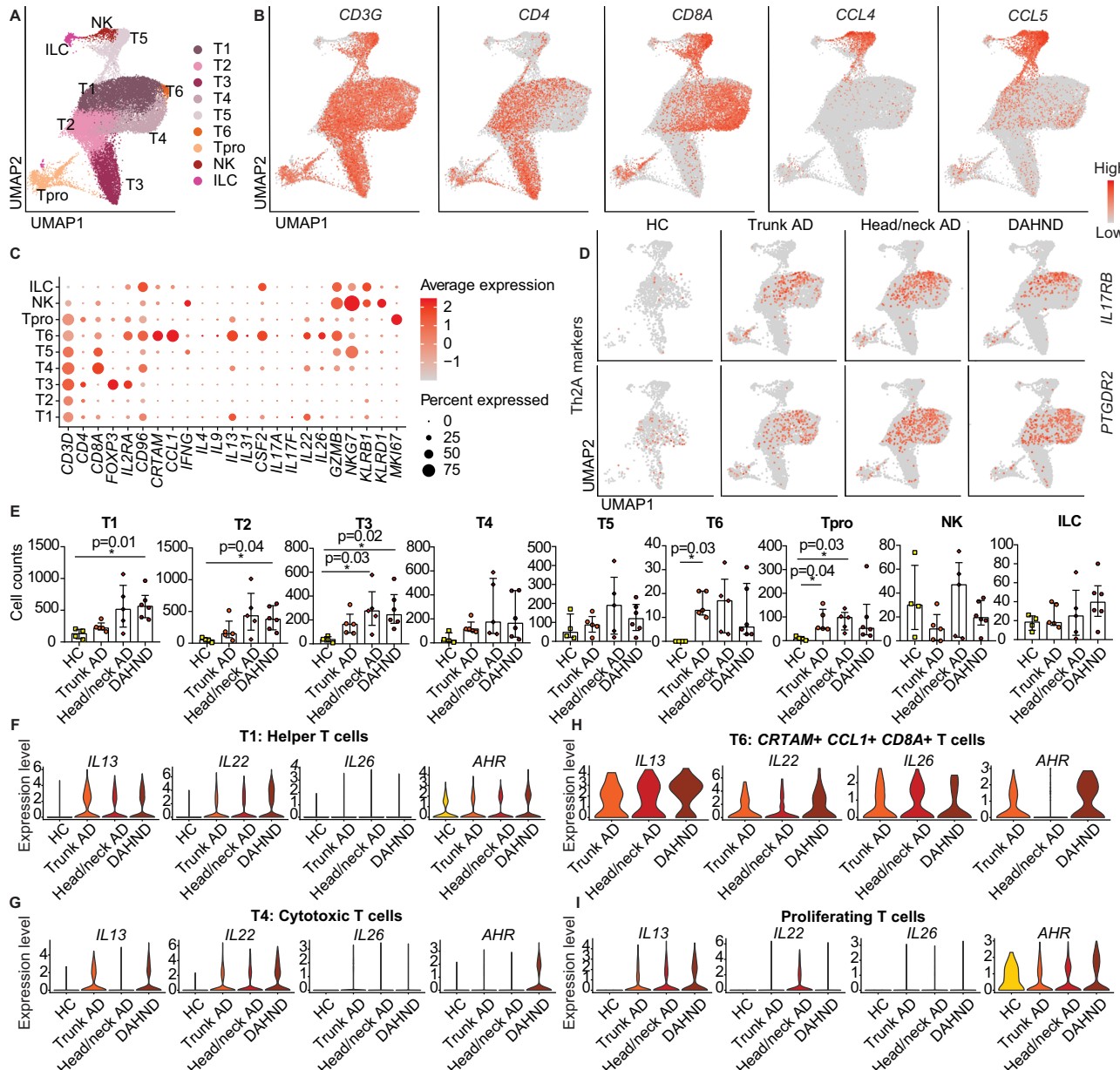

**Fig. 3 | Comparative analysis of lymphoid populations in DAHND, untreated head/neck and trunk AD, and HC samples. A** UMAP plot of the lymphoid cluster, consisting of T cells, NK cells, and ILCs. **B** Feature plots of selected T cell canonical genes. Intensity of expression levels for each cell is color-coded (red) and overlaid onto UMAP plots. **C** Dot plot with canonical markers indicative of respective lymphoid subsets. Coloring denotes expression levels within each cluster (red is high). **D** Feature plots of genes associated with so-called "Th2A" cells. Intensity of expression levels for each cell is color-coded (red) and overlaid onto UMAP plots. **E** Absolute cell counts per cluster across conditions: HC ($n = 4$), trunk AD ($n = 5$), head/neck AD ($n = 5$), and DAHND ($n = 6$); each dot represents a single individual; data are presented as median with IQR. Statistical significance was calculated using a Kruskal–Wallis test for multiple comparisons followed by a Dunn's post hoc test. **F–I** Violin plots of selected T cell lineage genes in T1, T4, T6, and Tpro clusters; y-axis indicates expression levels.

specifically expressed in keratinocytes and previously implicated in the inhibition of keratinocyte hyperplasia in an IL-23 inflammatory context[62], was significantly downregulated in DAHND compared to untreated head/neck AD (Fig. 7D). By contrast, *FGFBP1*, coding for fibroblast growth factor-binding protein FGF-BP that binds and activates FGF-1 and FGF-2[63], was strongly upregulated in DAHND (Fig. 7D).

Importantly, the components of the functional IL-22 receptor *IL22RA1* and *IL10RB* were induced in all AD groups compared to HC (Fig. 7F, G). Conversely, we did not detect upregulation of the IL-22 inhibitor *IL22RA2* in keratinocytes (Fig. 7F, G), further suggesting uninhibited IL-22 effects on the epidermal layer in DAHND[64]. Overall, keratinocyte responses were consistent with functionally relevant type

2 inhibition and enhanced type 22 immune activity within DAHND lesions.

**DAHND lesions lack proliferating endothelial cells, in contrast to untreated AD**

Among endothelial cell clusters, we were able to identify lymphatic (LEC), blood endothelial cells (BEC), and pericytes (P) (Fig. 8A, B). Numbers of venular blood endothelial cells BEC1, arteriolar BEC2, capillary BEC3, lymphatic endothelial cells (LEC) and pericytes (P) did not reveal significant differences across groups (Fig. 8C). While untreated trunk and head/neck AD showed significantly increased numbers of proliferating BECs, these were essentially absent not only in

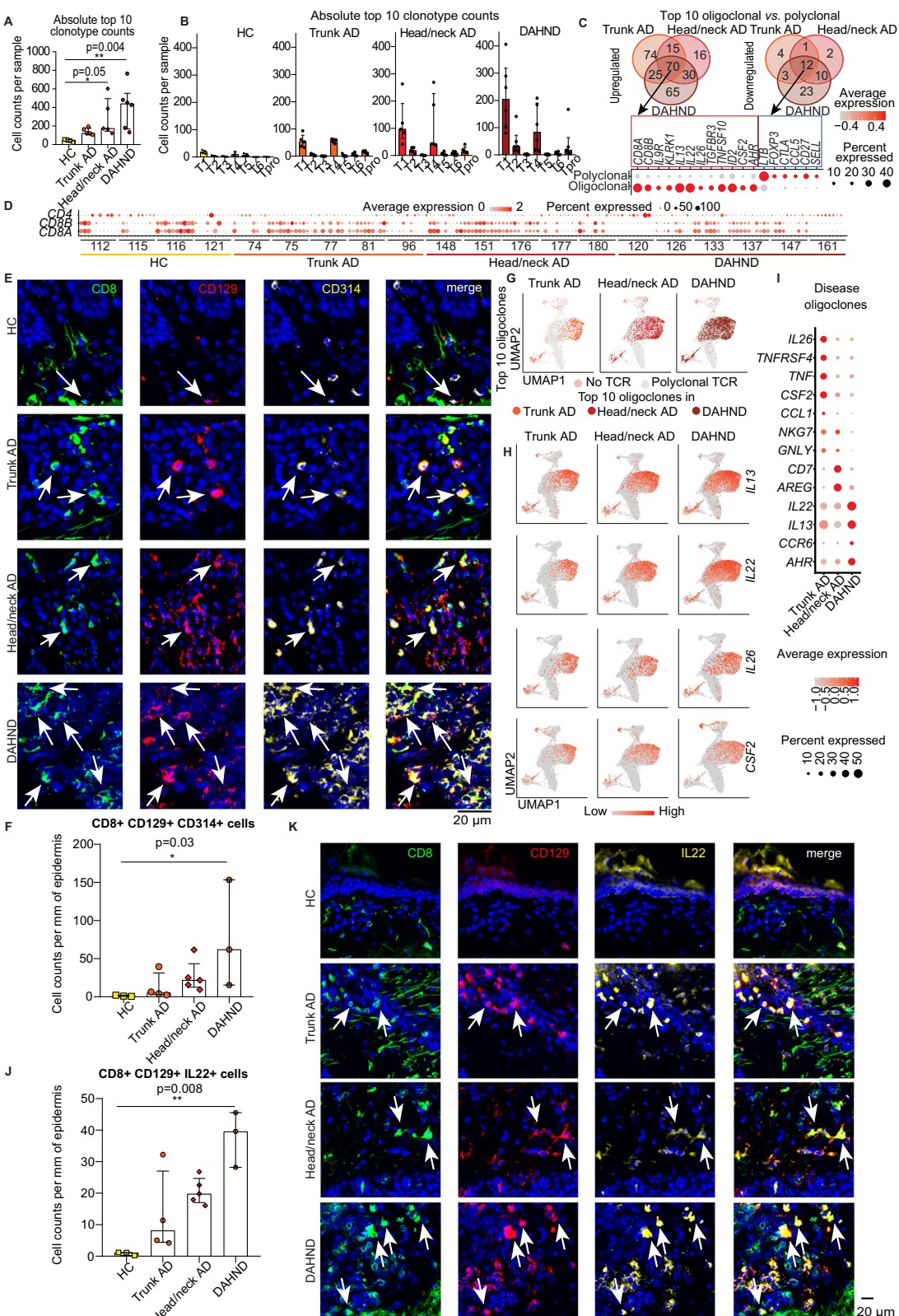

HC, but also in DAHND samples (Fig. 8C). Interestingly, the pro-inflammatory chemokine *CXCL12*[65], a ligand of CXCR4 which is currently being investigated as a therapeutic target in AD[66], was upregulated in both untreated head/neck AD and DAHND arteriolar BEC2s (Fig. 8D, Supplementary Data 4). However, in DAHND, we also found decreases in *VWF* and *FN1* gene expression (Fig. 8D, E), consistent with diminished

angiogenic tissue properties and decreased endothelial dysfunction, respectively[67–69]. We also found lower expression of *CCL14* in DAHND lymphatic endothelial cells (Fig. 8E), a chemokine previously associated with pathological neovascularization[70]. Taken together, these data suggest that angiogenic processes are likely not key to DAHND pathogenesis, despite the strongly pro-inflammatory microenvironment.

**Fig. 4 | Clonal landscape within disease groups. (A)** Total cell counts of the top 10 T cell receptor (TCR) clonotypes summarized for each condition: HC ($n = 4$), trunk AD ($n = 5$), head/neck AD ($n = 5$) and DAHND ($n = 6$); each dot represents a single individual; data are presented as median with IQR. **B** Absolute numbers of cells harboring the top 10 TCR clonotypes within each donor as distributed across lymphocyte clusters; data are presented as median with IQR. **C** Venn diagram of genes differentially up- or downregulated between the top 10 T cell oligoclones and the polyclonal T cell infiltrate, as calculated for each disease group, with expression levels of selected markers within the top 10 expanded clones (oligoclonal) and all other TCR+ cells (polyclonal). Differential gene expression was defined as log2 fold change >| ± 0.25| and adjusted $p < 0.05$. Selected genes are presented in a dot plot; coloring denotes expression levels within each group (red is high). **D** Expression of *CD8A, CD8B,* and *CD4* by each of the top 10 expanded T cell clones per sample; coloring denotes expression levels (red is high). **E, F** Representative pictures of multicolor immunofluorescence stainings for CD8, CD129 (*IL9R*), CD314 (*KLRK1*), with triple-positive cells appearing in white (arrows), as well as absolute cell counts

per mm of epidermis across conditions: HC ($n = 3$), trunk AD ($n = 4$), head/neck AD ($n = 5$) and DAHND ($n = 3$); each dot represents a single donor; data are presented as median with IQR. **G** Location of top 10 expanded T cell clones in UMAP plots of indicated disease groups. **H** Feature plots of T cell cytokine gene expression in each disease group. Intensity of expression levels for each cell is color-coded (red) and overlaid onto UMAP plots. **I** Gene expression dot plots displaying average and frequency of expression of selected markers for each disease group. Circle size represents the percentage of cells expressing the specific marker within a cluster. Coloring denotes expression levels within each group (red is high).
**J, K** Representative pictures of multicolor immunofluorescence stainings for CD8, CD129 (*IL9R*), and IL-22, with triple-positive cells appearing in white (arrows), as well as quantification of absolute cell numbers per mm of epidermis across conditions: HC ($n = 3$), trunk AD ($n = 4$), head/neck AD ($n = 5$) and DAHND ($n = 3$); each dot represents a single donor; data are presented as median with IQR. Statistical significance in this figure was calculated using a Kruskal–Wallis test for multiple comparisons followed by a Dunn's post hoc test.

## DAHND lesions show downregulation of AD-typical markers *COL6A5* and *COL6A6* in papillary and inflammatory fibroblasts

Confirming their remarkable heterogeneity, we identified 7 different fibroblast populations (FB1-FB6, FBpro) and 2 clusters of smooth muscle cells/myofibroblasts (SMC1, SMC2) (Fig. 9A). FB1 constituted the largest cluster harboring *CCN5+ SLPI+ DPP4+* cells, a phenotype most consistent with secretory-reticular fibroblasts[71] (Fig. 9B, Supplementary Data 3). We found two pro-inflammatory subsets, FB2 and FB4, that both characteristically expressed apolipoprotein E *APOE*, the neutrophil chemoattractant *CXCL2* and the fibulin family member of extracellular matrix glycoproteins *EFEMP1*[71], while only FB4 showed high expression of the chemokine ligand *CCL19* (Fig. 9B). We also detected *APCDD1+ COL18A1+* secretory-papillary FB3 and smaller mesenchymal subsets[72] of either cartilage component *COL11A1+* (FB5) or bone component *TNN+ SFRP1+ COL24A1+* (FB6) (Fig. 9B, Supplementary Data 3). While these populations were generally found across all patient groups, a smaller cluster of *MKI67+* proliferating fibroblasts was absent in healthy controls (Fig. 1C, Supplementary Data 2).

When we directly compared DAHND fibroblasts with location-matched untreated head/neck AD (Fig. 9C–F, Supplementary Data 5), we found strong downregulation of the AD-typical fibroblast markers[73] *COL6A5* and *COL6A6* particularly in FB3 and FB4 clusters of DAHND lesions (Fig. 9E, F, Supplementary Data 5), suggesting their induction by type 2 cytokines rather than constitutive expression in AD. Other mediators previously associated with type 2 inflammation[7,73,74] that were downregulated in DAHND, included the extracellular matrix component *POSTN* in FB3 and FB4, the chemokine ligands *CCL13* and *CCL26* in FB4, and the tenascin C gene *TNC* in FB1 and FB3 (Fig. 9C–F, Supplementary Data 5). Beyond these classically type 2 regulated mediators, we also found downregulation of various other inflammatory components in DAHND, such as chemokine ligands *CXCL1, CXCL2*, and *CXCL3* in FB3, the apolipoprotein genes *APOD* in FB2 and FB4 and *APOE* in FB1 and FB2, and the membrane alanyl aminopeptidase *ANPEP* in FB3 (Fig. 9C–F, Supplementary Data 5). Nevertheless, DAHND fibroblasts also showed upregulation of several other mediators. FB3 harbored increased levels of the anti-angiogenic chemokine *CXCL14*, consistent with decreased endothelial proliferation in DAHND (Fig. 9E, Supplementary Data 5). FB1 and FB2 showed strong upregulation of *CD74*, the receptor for macrophage migration inhibitory factor and previously implicated as key factor for the development of contact hypersensitivity (Fig. 9C, D, Supplementary Data 5)[75]. DAHND FB2 harbored increased expression levels of the CCR7 ligand *CCL19*, and FB4 upregulation of the alarmin *IL33* (Fig. 9D, F, Supplementary Data 5). Furthermore, FB3 in DAHND showed upregulation of the *BGN* gene coding for biglycan, a proteoglycan also previously implicated in contact hypersensitivity reactions (Fig. 9E, Supplementary Data 5)[76].

Whereas both SMC clusters harbored cytoskeleton-specific markers such as α-smooth muscle actin (*ACTA2*), transgelin (*TAGLN)* and

myosin light chain 9 (*MYL9*) among their top DEGs, the smaller SMC2 cluster additionally demonstrated an upregulation of inflammatory cytokines such as *CCL26* and *CCL19* (Fig. 9B, Supplementary Data 3). While SMC2 showed upregulation of the AD signature chemokine *CCL26* as well as *POSTN* only in untreated head/neck and trunk AD, *CCL19*, a ligand of CCR7, was more abundantly detected in DAHND lesions, with concomitant increases in *CCL8* expression (Fig. 9G, Supplementary Data 4), which may shape the specific inflammatory milieu in the dermis of these lesions. Interestingly, the CCR7 ligand *CCL21*, that has previously been demonstrated to be a key contributor to Th17/Th22 chemotaxis[77], was upregulated in DAHND SMC2 (Fig. 9G). In summary, we found active inhibition of type 2 inflammation in stromal cells of the dermis, with upregulation of specific pro-inflammatory mediators potentially involved in DAHND pathogenesis.

## Discussion

Dupilumab, a therapeutic monoclonal IgG4 antibody, functions in treating AD by binding the IL-4 receptor alpha chain (IL-4Rα), a regulator of type 2 T cell differentiation and a mediator of IL-4/IL-13 cytokine signaling[78–80]. Dupilumab has been shown to downregulate expression of over 800 genes implicated in AD, thereby mitigating type 2 chemokines and the associated inflammation[7,8,81]. Nevertheless, dupilumab treatment has been associated with secondary adverse effects, such as psoriasiform dermatitis, arthritis, rosacea, and conjunctivitis/blepharitis, which introduces the idea that suppression of the Th2/Tc2 pathway may lead to compensatory Th1-, Th17-, and Th22 directed responses[78,82–85]. Indeed, inflammatory ocular manifestations are believed to arise from heightened Th1/Th17 activation, leading to increased expression of IL-17, *Demodex* colonization, and a subsequent impairment of Meibomian gland function[86]. Similar mechanisms have been proposed for the development of rosacea-like folliculitis[87]. However, reasons for a regional preference of the head/neck area for these manifestations remain unclear. Location-dependent differences in skin homeostasis have mostly been investigated in relation to skin lipid composition and the microbiome[88–90], with studies showing that microbial diversity depends more strongly on the body site than on the individual[91]. However, investigations of site-specific immune activation in atopic individuals have yet to be performed. In our study, we found that AD from the trunk and from the head/neck area were different in a limited number of inflammatory mediators. Blood endothelial cells from the head/neck area showed increased expression levels of the strongly pro-inflammatory chemokine *CXCL12*, a ligand of CXCR4 currently assessed as a treatment target in AD[66]. Dendritic cells from trunk AD showed decreases in *IL1B*, while the inflammatory proteinase *MMP12* was significantly increased, suggesting some nuanced differences in innate immune regulation depending on the body site in AD. We also found significantly increased numbers of cells of the B cell lineage in DAHND lesions. Previously, disease severity and progression

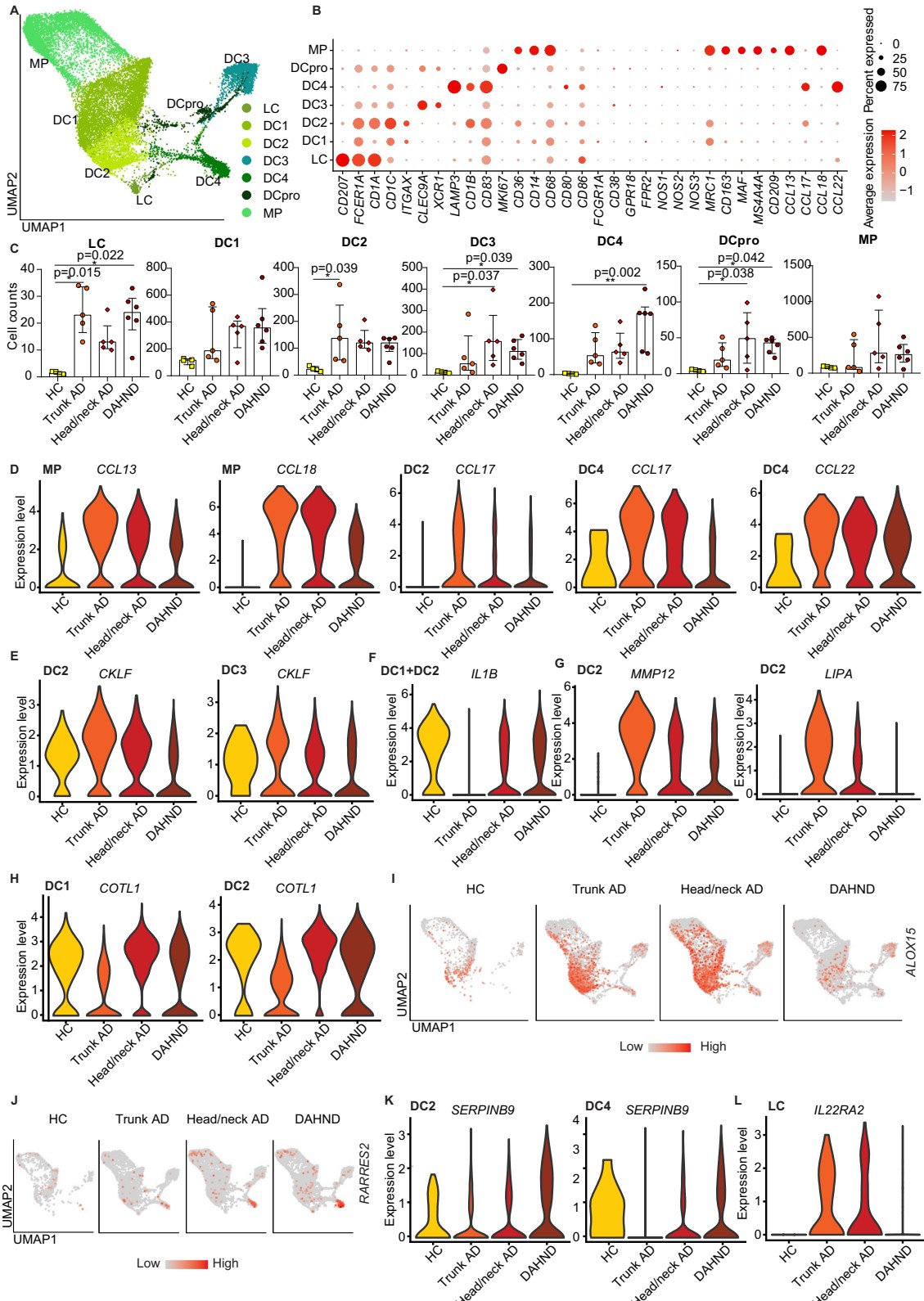

**Fig. 5 | Characterization of myeloid cells in DAHND, untreated head/neck and trunk AD, as well as HC samples. A** UMAP plots of myeloid cell subsets. **B** Dot plot of canonical markers for respective dendritic cell (DC), Langerhans cell (LC), and macrophage (MP) subsets; coloring denotes expression levels within each cluster (red is high). **C** Absolute cell counts per cluster across conditions: HC (*n* = 4), trunk AD (*n* = 5), head/neck AD (*n* = 5), and DAHND (*n* = 6); each dot represents a single donor; data are presented as median with IQR. Statistical significance was calculated using a Kruskal–Wallis test for multiple comparisons followed by a Dunn's post hoc test. **D–H** Violin plots of selected genes in myeloid cell clusters; y-axis indicates expression levels. **I, J** Feature plots of *ALOX15* and *RARRES2* expression in each disease group. Intensity of expression levels for each cell is color-coded (red) and overlaid onto UMAP plots. **K, L** Violin plots of selected genes in DC2, DC4, and LC clusters.

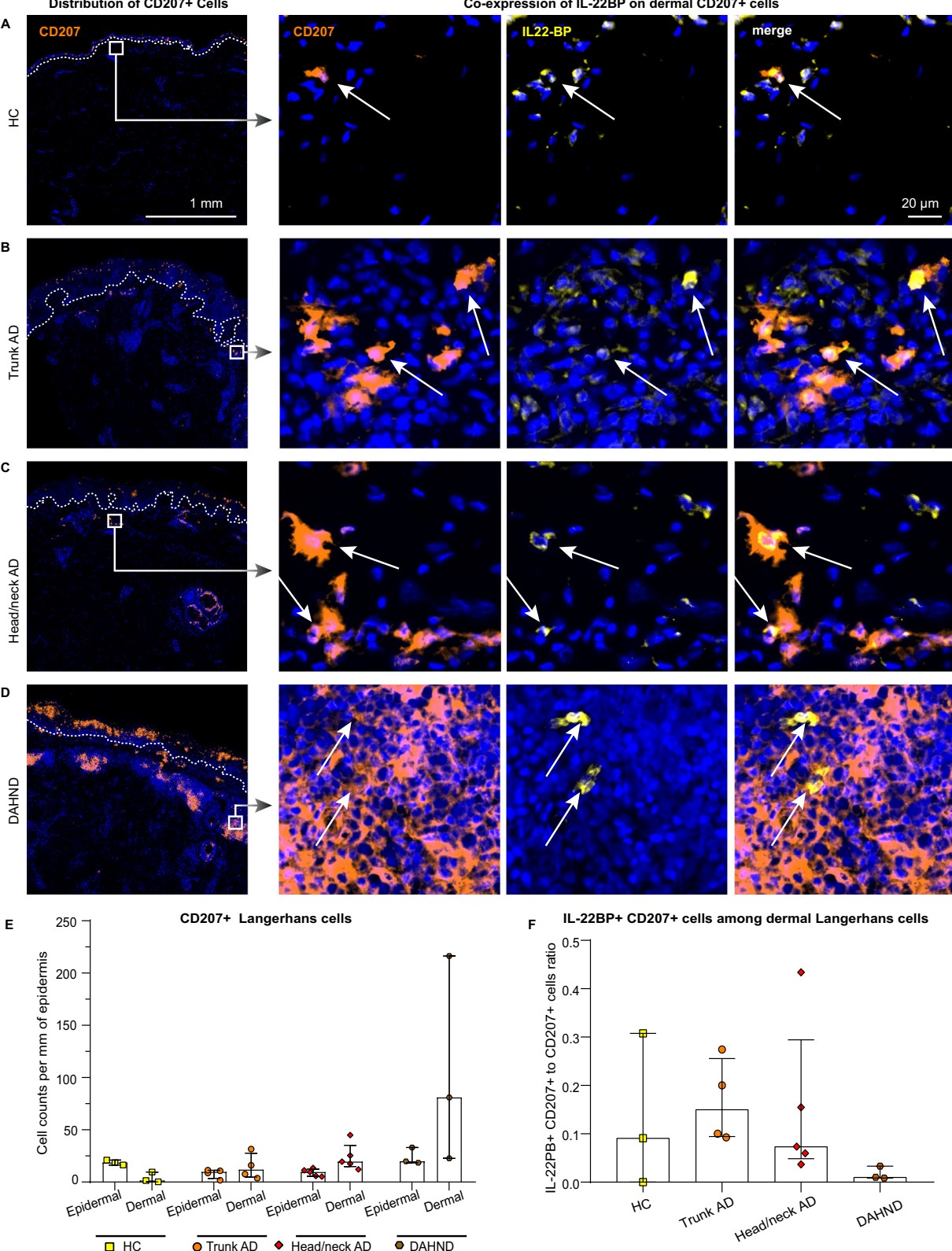

**Fig. 6 | Langerhans cells express the IL22 inhibitor IL-22BP. A–D** Representative immunofluorescence images of CD207 and IL-22BP (*IL22RA2*) in HC skin, trunk AD, head/neck AD, and DAHND; dotted lines in left images denote dermo-epidermal junction. **E** Quantification of epidermal and dermal CD207+ Langerhans cells; absolute cell counts are given per mm of epidermis. HC (*n* = 3), trunk AD (*n* = 4), head/neck AD (*n* = 5), and DAHND (*n* = 3); each dot represents a single donor; data are presented as median with IQR. **F** Ratio of IL-22BP+ CD207+ cells vs. all CD207+ Langerhans cells within the dermis in respective disease groups: HC (*n* = 3), trunk AD (*n* = 4), head/neck AD (*n* = 5), and DAHND (*n* = 3); each dot represents a single donor; data are presented as median with IQR.

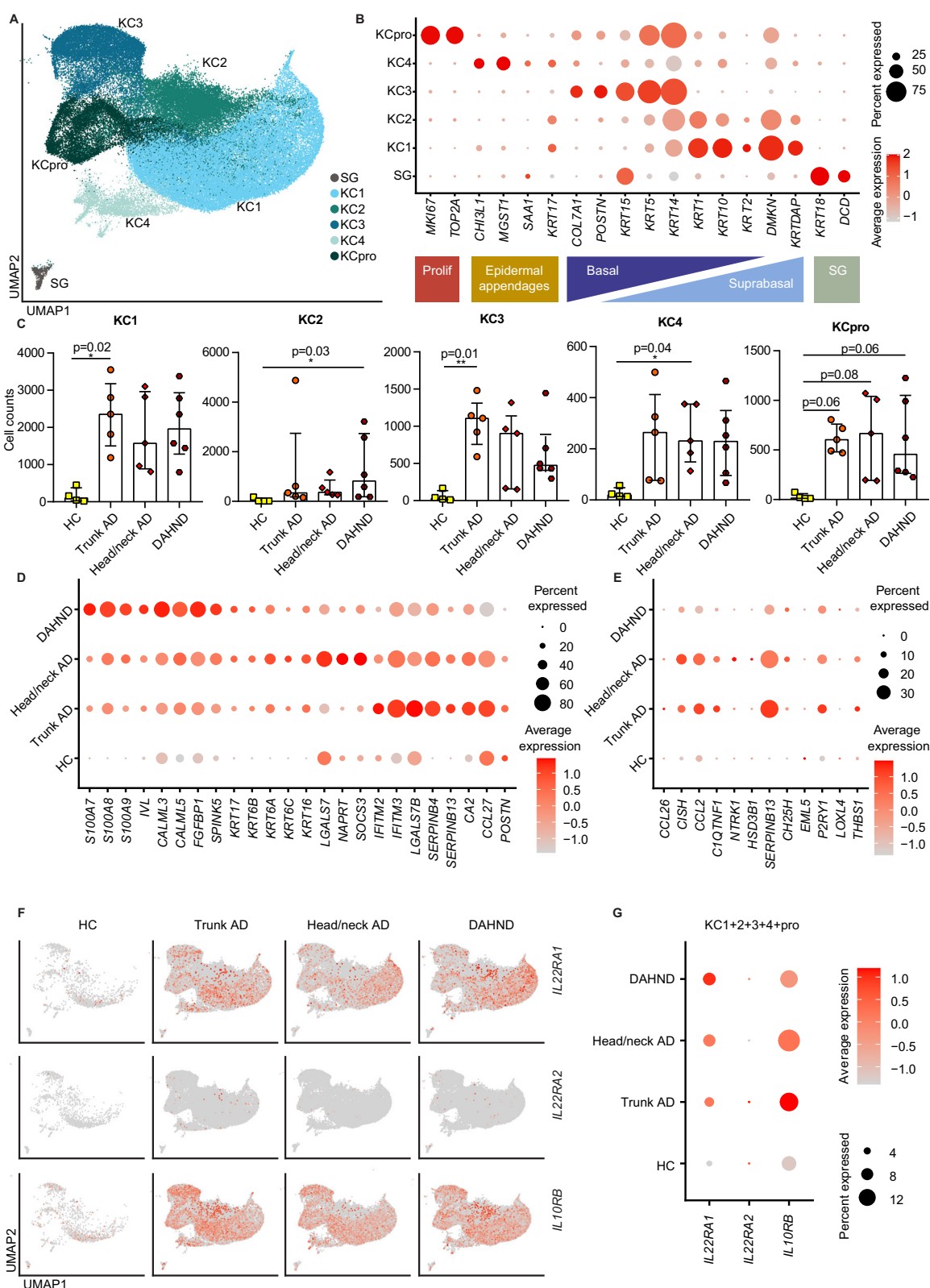

have been reported to be positively correlated with the number of skin-infiltrating B cells in a variety of diseases, such as pemphigus, scleroderma, as well as AD[92–94], suggesting a role in DAHND pathogenesis. Among lymphocytes, we found oligoclonally expanded T cells to be present in all forms of AD, but expansion was greatest in DAHND, with a pronounced *IL13/IL22/AHR/IL9R* signature that was corroborated in multicolor immunofluorescence as well as bulk RT-PCR

quantification in independent sample sets. In line with the understanding that the receptor for IL-22 (which is comprised of the receptor dimer *IL22RA1* and *IL10RB*) is expressed in keratinocytes, these cells exhibited pronounced hyperactivation, surpassing even the levels observed in untreated AD. This hyperactivation was made evident through the elevated expression of keratinocyte activation markers, namely *S100A7, S100A8*, and *S100A9*. Importantly, neither

**Fig. 7 | Comparative analysis of keratinocyte (KC) and sweat gland (SG) populations in DAHND, untreated head/neck and trunk AD, and HC samples.** **A** UMAP plot of KC and SG subsets. **B** Dot plot of canonical markers for respective epithelial subsets; coloring denotes expression levels within each cluster (red is high). **C** Absolute cell counts per KC cluster across conditions: HC ($n$ = 4), trunk AD ($n$ = 5), head/neck AD ($n$ = 5), and DAHND ($n$ = 6); each dot represents a single individual; data are presented as median with IQR. Statistical significance was calculated using a Kruskal−Wallis test for multiple comparisons followed by a Dunn's post hoc test. **D** Dot plot of selected genes showing the differences between disease groups in all keratinocytes (KC1, KC2, KC3, KC4, KCpro) combined; coloring

denotes expression levels within each group (red is high). **E** Dot plot of IL-13-associated keratinocyte response genes from a published dataset[59] showing differences between disease groups in all keratinocytes (KC1, KC2, KC3, KC4, KCpro) combined; coloring denotes expression levels within each group (red is high). **F** Feature plots of IL22-receptor components as expressed in epithelial cells of each disease group. Intensity of expression levels for each cell is color-coded (red) and overlaid onto UMAP plots. **G** Dot dot plot of IL22-receptor components showing the differences between conditions for all keratinocytes combined; coloring denotes expression levels within each group (red is high).

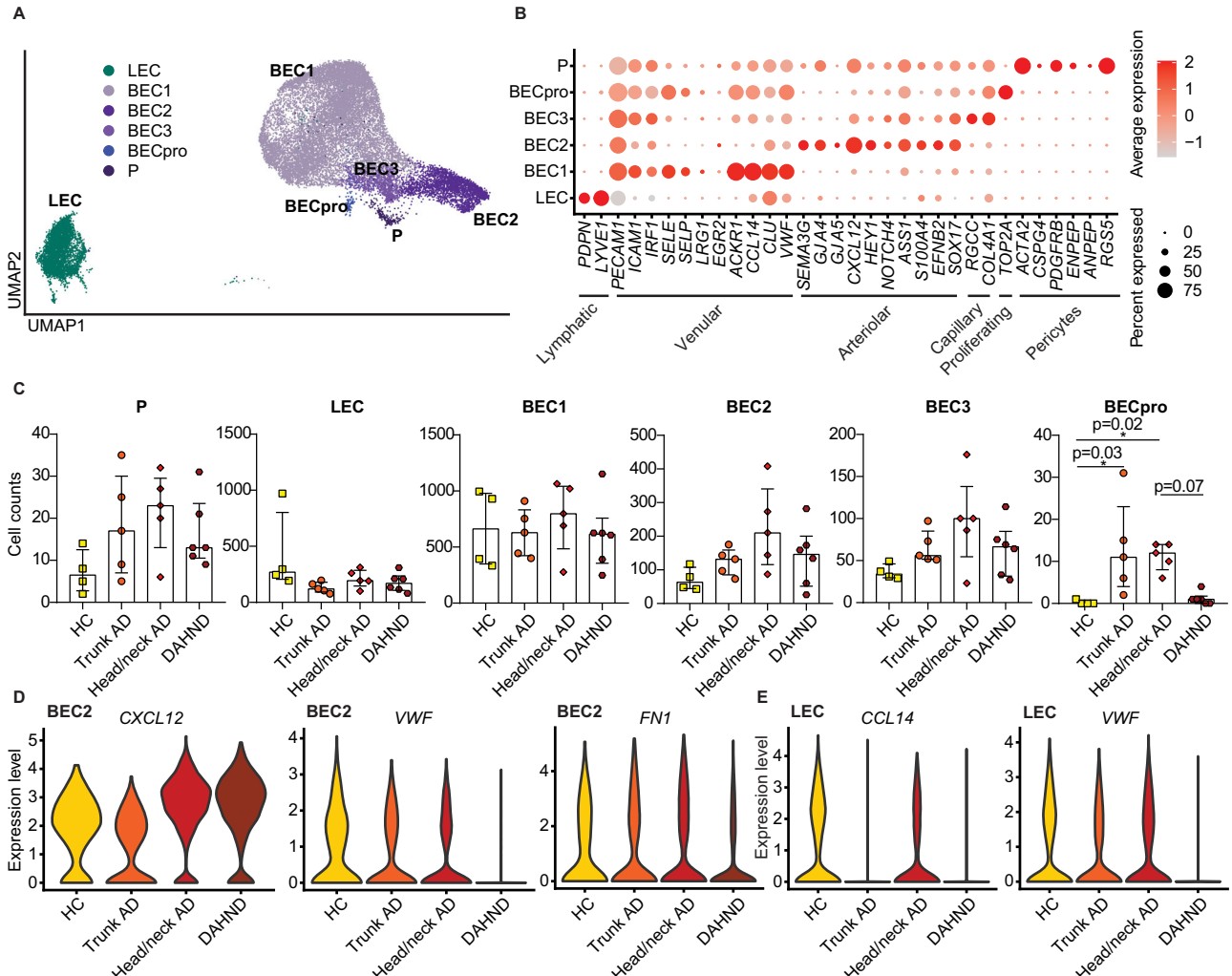

**Fig. 8 | Comparative analysis of endothelial cell populations in DAHND, untreated head/neck and trunk AD, as well as HC samples.** **A** UMAP plot of endothelial cell subsets. **B** Dot plot of canonical cell type markers of endothelial cell subpopulations; coloring denotes expression levels within each cluster (red is high). **C** Absolute cell counts of respective clusters across conditions: HC ($n$ = 4),

trunk AD ($n$ = 5), head/neck AD ($n$ = 5), and DAHND ($n$ = 6); each dot represents a single individual; data are presented as median with IQR. Statistical significance was calculated using a Kruskal−Wallis test for multiple comparisons followed by a Dunn's post hoc test. BEC blood endothelial cells; LEC lymphatic endothelial cells; P pericytes. (**D**, **E**) Violin plots of selected genes in arteriolar BEC2 and LEC clusters.

keratinocytes, nor the majority of Langerhans cells, expressed the IL-22 inhibitor *IL22RA2* (IL-22BP) in DAHND, suggesting uninhibited effects of type 22 inflammation in these lesions. Conversely, type 2 inflammation was effectively blocked in DAHND despite ongoing IL-13 expression in T-cells, as evidenced by decreased expression of downstream mediators, which is in line with the fact that these lesions generally lack significant spongiosis[24], a type 2-induced hallmark of eczematous skin reaction patterns[95,96]. Despite previous associations of IL-22 with type 17 immune activation[97], *IL17A* and *IL17F* transcripts were largely absent in DAHND.

Not only epidermal, but also dermal cells showed a distinct inflammatory micromilieu in DAHND when compared to untreated AD. Secretory papillary DAHND fibroblasts expressed higher levels of the anti-angiogenic mediator *CXCL14*, consistent with the absence of endothelial cell proliferation in DAHND in contrast to untreated AD. Interestingly, DAHND fibroblasts overexpressed several mediators previously implicated in contact hypersensitivity such as *CD74*[75] and *BGN*[76]. In line, some authors regard DAHND as a variant of persistent allergic contact dermatitis to fragrances or other haptens applied to the head/neck area[18,31].

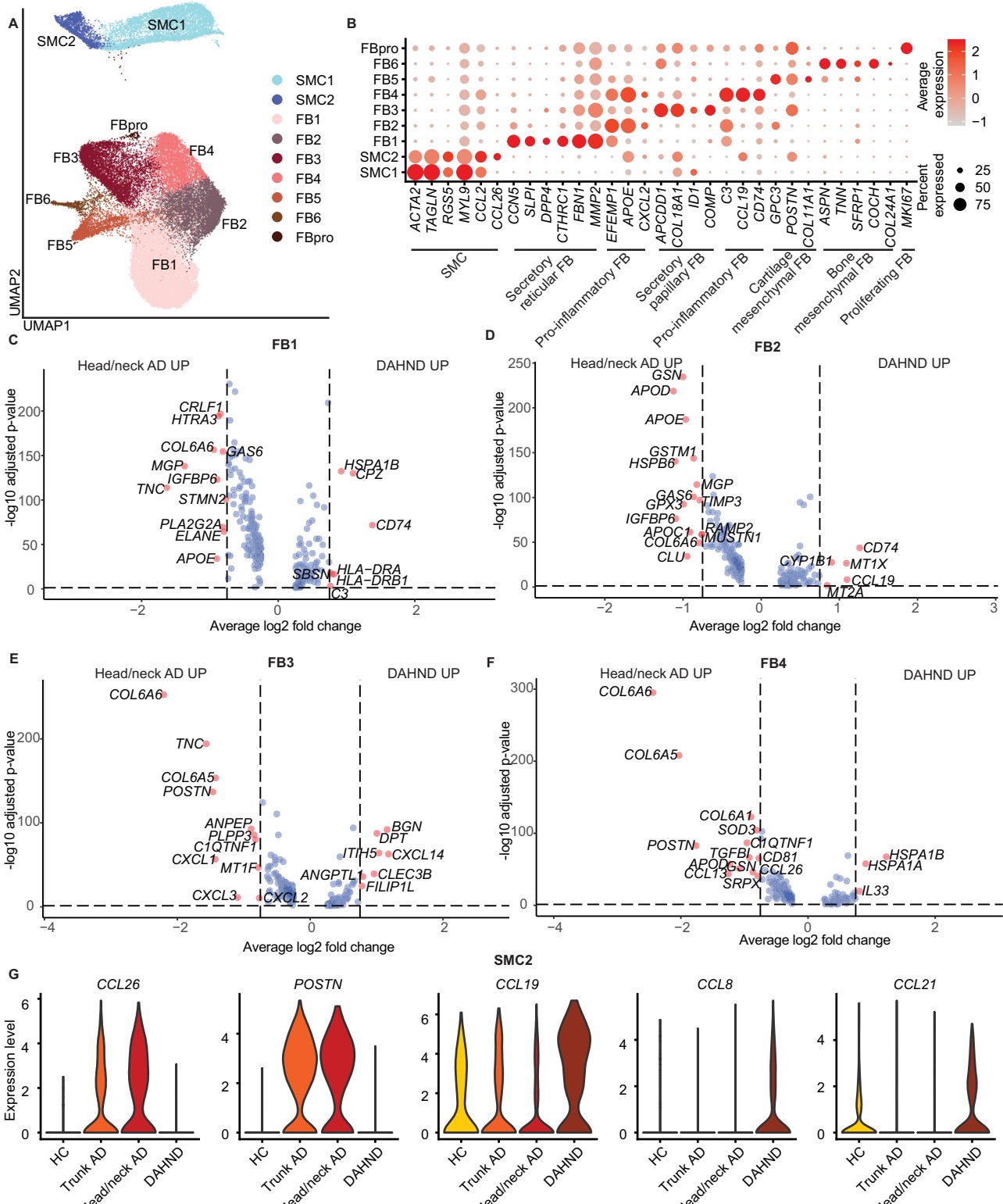

**Fig. 9 | Comparative analysis of fibroblast (FB) and smooth muscle cell (SMC) populations in DAHND, untreated head/neck and trunk AD, and HC samples.** **A** UMAP plot of SMC and FB cell subsets. **B** Dot plot of canonical cell type markers of FB and SMC; coloring denotes expression levels within each cluster (red is high). **C–F** Volcano plots of differentially expressed genes between DAHND and untreated head/neck AD in FB1, FB2, FB3, and FB4 based on Supplementary Data 5

for respective FB clusters, i.e., DEGs were defined as log2 fold change >| ± 0.25| and adjusted $p < 0.05$ using a two-sided Wilcoxon rank-sum test and Bonferroni correction; x-axis indicates average log fold change, y-axis indicates the −log10 of the adjusted $p$ value. **G** Violin plots of selected genes depicting differences between conditions in the SMC2 subpopulation.

Besides classical haptens, commensals such as *Malassezia* or *Demodex spp.* may also play a major role in the induction of DAHND. Increased baseline *Malassezia*-specific IgE has been associated with the appearance of DAHND in AD patients[26–29,98,99]. The *Malassezia sympoidalis*-derived allergen Mala s13 has great homology to stress-induced human thioredoxin and may thereby function as an auto-reactive T-cell antigen[100]. In line, AD skin has been demonstrated to harbor Mala s13-reactive T-cells that can contribute to skin inflammation[100]. In addition, recent experiments in *Demodex* infected IL4ra-/- mice have demonstrated that the absence of type 2 immunity can lead to *Demodex* outgrowth and aberrant cutaneous inflammation[101]. Consistently, patients with severe *Demodex*-associated skin diseases such as rhinophyma show increased expression levels of IL-22 and IFN-γ along with decreased IL-13 levels within the skin[101]. As *Malassezia* and *Demodex spp.* are predominantly found in the human head/neck area, this observation could explain the regional restriction of DAHND in affected patients. Importantly, IL-22 has been previously linked to both antimicrobial responses and contact hypersensitivity[102]. It is therefore tempting to speculate that DAHND reflects an IL-22 dominated hypersensitivity reaction to commensal colonization in the absence of functioning IL-13 signaling. Oligoclonal, and thus potentially antigen-specific expansion of *IL22*-producing *AHR* + T cells may therefore be a result of aberrant or exaggerated antimicrobial defense mechanisms, leading to the clinical appearance of a hypersensitivity rash in AD patients who develop DAHND. These data might also suggest that in addition to its pathogenic role in atopic diseases, type 2 inflammation could have a regulatory impact on other T cell arms, including type 22 cells. While this study was limited by an overall small sample size, the consistent dysregulation of type 22-associated inflammation in DAHND might be relevant for future treatment approaches.

## Methods

### Patient recruitment and sample processing

After obtaining written informed consent from all study patients, samples were collected under an approved protocol by the Ethics Committee of the Medical University of Vienna, Austria (EK 1360/2018) and subjected to different methods of analysis (Supplementary Data 1). Patients were compensated for providing skin punch biopsies. Patients were recruited irrespective of sex/gender (self-identified) due to limited patient numbers available. No sex/gender-specific analyses were performed due to a small sample size. Full-thickness skin punch biopsies were taken and either snap-frozen for immunohistochemistry and quantitative RT-PCR or immediately processed for single-cell suspensions by using the Whole Skin Dissociation Kit, human (130-101-540) by Miltenyi Biotech (Bergisch Gladbach, Germany), according to the manufacturer's instructions[103]. Cells were then immediately subjected to scRNA-seq processing using the Chromium Next GEM Single Cell 5′ Library & Gel Bead Kit v1.1 (PN-1000165), Chromium Single Cell 5′ Library Construction Kit (PN-1000020), Chromium Single Cell V(D)J Enrichment Kit, Human T Cell (PN-1000005), Chromium Next GEM Chip G Single Cell Kit (PN-1000120), and Single Index Kit T Set A (PN-1000213) (10X Genomics, Pleasanton, CA), according to the manufacturer's protocol. After library construction and appropriate quality control measures, pooled samples were subjected to sequencing using the Illumina NovaSeq platform and the 150 bp paired-end configuration.

### Analysis of scRNA-seq data

For scRNA-seq analyses, Seurat (v. 4.0.4 - 4.4.0)[104] and scran packages v. 1.26.2[105] as well as Bioconductor packages (BiocManager version 3.16) SingleCellExperiment version 1.20.1 and EnhancedVolcano version 1.16.0 were used in R 4.2.2 (2022-10-31). R Bioconductor package scDblFinder (version 1.10.0)[106] was used for doublet removal. The command "computeDoubletDensity" with 200 nearest neighbors was applied to calculate the doublet score for each cell. The score was log10 of the ratio between simulated doublet and total number of neighbors, and all cells with a score higher than 3 times the median absolute deviation were discarded. CDR3 amino acid sequences derived from TCR sequencing were added to corresponding cells based on their barcodes. Cells containing a high percentage of mitochondrial genes (>10%), and either very low (<500) or high numbers (in the range of 3000 to 7000, evaluated on an individual donor-basis) of unique genes (nFeature_RNA) were filtered out. Samples were processed using fast integration by the reciprocal PCA (RPCA) workflow (https://github.com/satijalab/seurat/blob/master/vignettes/integration_rpca.Rmd)[104]. Due to a high ambient RNA "noise" originating from lysed keratinocytes, we performed a two-step analysis. Log-normalization of the dataset and identification of 2000 highly variable features were followed by a principal component analysis and unsupervised clustering with the Louvain algorithm at a resolution of 0.5 and 50 dimensions. Uniform Manifold Approximation and Projection (UMAP)[107] allowed visualization of cell clusters in a two-dimensional space. Thereafter, we identified certain keratinocyte/fibroblast-specific genes uniformly present in *PTPRC*+ clusters due to ambient RNA contamination, e.g., in the *CD3D*+ T cell population. Since the expression levels of these genes formed a bimodal distribution between keratinocyte/fibroblast clusters and other cell types, we were able to define clear expression level cutoffs that represented ambient RNA contamination. These genes and corresponding cutoffs were: expression level >2 for the genes *S100A13*, *S100A16*; expression level >2.5 for the genes *S100A14*, *GSTP1*, *GSN*; expression level >3 for the genes *LGALS7*, *DSP*, *S100A4*, *S100A10*, *KRT2*, *LY6D*, *KRT6B*, *KRT6C*, *FABP5*, *KRT17*, *DMKN*, *AQP3*, *CXCL14*; expression level >3.5 for the genes *S100A2*, *SFN* and *PERP;* and expression level >4 for the genes *KRT1*, *KRT10*, *KRT5*, *KRT14*, *S100A6*, *S100A7*, *S100A8*, *S100A9*, *KRT6A*, *KRT16*, *KRTDAP*, *LGALS7B*. Upon amending the data, fast integration using the reciprocal PCA workflow was conducted again with the same parameters. Canonical markers of cell subsets were visualized in FeaturePlots and cell subpopulations were then identified by manual adjustment of the resolution in the "FindClusters" function. Variations in cell numbers were evaluated using a Kruskal–Wallis test followed by a Dunn's post hoc test in GraphPad Prism software (v. 8.0.1 – 10.1.1) (GraphPad Software, La Jolla, Calif). Markers of defined cell subpopulations and differentially expressed genes between sample groups were calculated using the "FindAllMarkers" command applying the Wilcoxon rank-sum test as well as the following parameters: adjusted *p*-values of <0.05, log-fold change >0.25, Bonferroni correction for multiple comparisons, default min.pct value of 0.01, if not indicated otherwise. For final display of the data, ribosomal and mitochondrial protein genes were filtered out.

### Multicolor skin immunofluorescence staining and image acquisition

4–6-mm skin punch biopsies were embedded in optimal cutting temperature compound (O.C.T., Tissue-Plus, Scigen Scientific, Gardena, Calif) and stored at −80 °C until further processing[44,108]. Biopsy sections were cut and mounted on SuperFrost Plus adhesion slides (Thermo Fisher Scientific, Waltham, Mass), air dried and fixed in ice-cold acetone (Sigma-Aldrich, St Louis, MO) for 10 min. Each staining step was preceded by washing with PBS and blocking with PBS containing 2% BSA (Sigma-Aldrich), 2% mouse or goat serum (Dako, Jena, Germany) and Human TruStain FcX (Fc Receptor Blocking Solution) (Biolegend, San Diego, Calif) for 20–30 min in a humid chamber at room temperature (RT), to minimize background staining. Slides were stained with the following primary mouse antibodies: AF647-conjugated anti-CD79a (1:50, clone: HM47, Cat: 333516, Lot: B313857, Biolegend), or a combination of PE/Dazzle 594-conjugated anti-CD207 (Langerin) (1:20, clone: 4C7, Cat: 144212, Lot: B362308, Biolegend) and AF647-conjugated

anti-IL-22BP (1:20, clone: # 875504, Cat: FAB10871R-100UG, Lot: 1713824, R&D Systems). For the triple stainings, the slides were incubated with primary PE-conjugated anti-CD129 (*IL-9R*) (1:30, clone: AH9R7, Cat: 310404, Lot: B340453, Biolegend) antibody, followed by AF546-conjugated goat-anti-mouse IgG (1:400, Cat: A11030, Lot: 2026145, Thermo Fisher Scientific) second-step. After washing and blocking as described above, additional combinations of antibodies were applied: AF488-conjugated anti-CD8 (1:10, clone: SK1, Cat: 344716, Lot: B351669, Biolegend) together with either AF647-conjugated anti-CD314 (*KLRK1*) (1:20, clone: 1D11, Cat: 320826, Lot: B360539, Biolegend) or AF647-conjugated anti-IL22 (1:20, clone: MH22B2, Cat: 567160, Lot: 3101605, BD Biosciences). Cell nuclei were counterstained with 1 μg/ml 4,6-diamidino-2-phenylindole dihydrochloride (DAPI) (Roche Diagnostics, Mannheim, Germany) and placed in aqueous mounting medium PermaFluor (Thermo Fisher Scientific). Corresponding isotype controls, such as AF647-conjugated mouse IgG1 (1:10, MOPC-21, Cat: 400136, Lot: B287199, Biolegend), AF488-conjugated mouse IgG1 (1:20, MOPC-21, Cat: 400129, Lot: B354284, Biolegend), PE-conjugated mouse IgG2b (1:10, MG2b-57, Cat: 401208, Lot: B353563, Biolegend), AF647-conjugated mouse IgG2a (1:2.5, MOPC-173, Cat: 400234, Lot: B356813, Biolegend), AF594-conjugated mouse IgG2a (1:50, MOPC-173, Cat: 400280, Lot: B331958, Biolegend), and AF647-conjugated mouse IgG2b (1:50, MPC-11, Cat: 400330, Lot: B379318, Biolegend), were used to confirm staining specificity. Pictures were taken using the TissueFAXS v6 imaging system (TissueGnostics GmbH, Vienna, Austria), equipped with a Zeiss Axio Observer Z1 microscope (Carl Zeiss Inc, Jena, Germany), Zeiss Plan-Neofluar objectives (primary objective 20x/0.5, ocular objective 10x), Spectra III 8-LCR-XN light engine, a PCO PixelFly monochrome camera (PCO, Kelheim, Germany) and Pixelink PL-623 color camera (Pixelink, Rochester, NY). Cell numbers were quantified using TissueFAXS v6 (TissueGnostics GmbH, Vienna, Austria), as previously described[108,109].

### Quantitative RT-PCR

mRNA for quantitative RT-PCR was isolated from snap-frozen tissue using TRI reagent (Sigma-Aldrich), transcribed to cDNA, and analyzed using TaqMan Gene Expression Master Mix (4369016) and TaqMan Gene Expression Assays (Thermo Fisher Scientific), i.e., *IL4*, Hs00929862_m1; *IL5*, Hs00174200_m1; *IL9*, Hs00914237_m1; *IL10*, Hs00961622_m1; *IL13*, Hs99999038_m1; *IL31*, Hs01098710_m1; *IL17A*, Hs00174383_m1; *IL17F*, Hs00369400_m1; *IL22*, Hs01574154_m1; *IL26*, Hs00218189_m1; *IFNG*, Hs00174143_m1, according to the manufacturer's instructions[35].

### Statistics

Statistical significance of differences in cell counts of multicolor immunofluorescence or scRNA-seq was evaluated using Kruskal–Wallis test for multiple comparisons followed by a Dunn's post hoc test. Differential gene expression in scRNA-seq was calculated with the "FindAllMarkers" command using a two-sided Wilcoxon rank-sum test, with adjusted $p$-values of <0.05, log-fold change >0.25, Bonferroni correction for multiple comparisons, and default min.pct value of 0.1, if not otherwise indicated. For single-cell analyses, absolute cell counts per sample and cluster are listed in Supplementary Data 2 and the Source data file, and source data from quantitative RT-PCR and immunohistochemistry analyses displayed in figures are all shown in the Source data file.

### Reporting summary

Further information on research design is available in the Nature Portfolio Reporting Summary linked to this article.

## Data availability

The processed 10X Genomics datasets generated in this study have been deposited in the Gene Expression Omnibus (GEO) database under accession code GSE230575. Data from healthy control and trunk AD samples are equivalent to samples with the GEO accession numbers GSE173205 and GSE222840, respectively, from previously published datasets from our laboratory[44,103]. All other data are available in the article and its Supplementary files or from the corresponding author upon request. Source data are provided with this paper.

## Code availability

The code for the data generated in this study is provided in the Supplementary folder "Supplementary Software".

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

## Acknowledgements

We thank the Biomedical Sequencing Facility (BSF) at CeMM Research Center for Molecular Medicine of the Austrian Academy of Sciences, Vienna, for assistance with next-generation sequencing. This work was funded by a research grant to PMB from the LEO Foundation (grant number LF-OC-20-000621). S.C. was supported by the NCATS NRSA TL1 in Clinical & Translational Science (TL1TR004420) award.

## Author contributions

Conceptualization: C.B. and P.M.B.; methodology: N.A., M.A.M. and J.G.; investigation: N.A., M.P. and T.A.; visualization: N.A., S.C. and J.P.L.; funding acquisition: P.M.B.; project administration: C.B. and P.M.B.; supervision: CB and PMB.; writing—original draft: C.B., N.A. and P.M.B.; writing—review & editing: N.A., S.C., T.A., J.P.L., M.P., J.R., E.R.C., C.J., W.D. and J.G.

## Competing interests

C.B. has received personal fees from ALK, Mylan, LEO Pharma, Pfizer, Sanofi Genzyme, Eli Lilly, Novartis, Almirall and AbbVie. C.B. is an investigator for Novartis, Sanofi, Abbvie, Elli Lilly, LEO Pharma and Galderma. C.J. has received personal fees from LEO Pharma, Pfizer, Recordati Rare Diseases, Eli Lilly and Company, Novartis, Takeda, AbbVie, Janssen, UCB, Sandoz, Kyowa Kirin, Bristol Myers Squibb, and Almirall. C.J. is an investigator for Eli Lilly and Company, Novartis, Innate Pharma, and 4SC (grant paid to her institution). W.D. has received personal fees from Pfizer, Incite, Eli Lilly, TWI Biotechnology, Fresenius Kabi, Epiarx Diagnostics, and Boehringer Ingelheim. W.D. has received research support from Pfizer, Advanced Cell Diagnostics/Bio-techne, Abbvie, Bristol Myers Squibb, and Incite. W.D. receives licensing fees from EMD/Millipore/Sigma. JG has received personal fees from AbbVie and Novartis. P.M.B. has received personal fees from LEO Pharma, Pfizer, Sanofi Genzyme, Eli Lilly, Novartis, Celgene, UCB Pharma, Biotest, Boehringer Ingelheim, AbbVie, Amgen, Arena Pharmaceuticals, GSK, BMS, Almirall and Regeneron. PMB has received research support from Pfizer. N.A., S.C., T.A., J.P.L., M.P., M.A.M., J.R. and E.R.C. declare no competing interests.
