## [Peer Review File · Nature Communications]

Dupilumab-associated head and neck dermatitis shows a pronounced type 22 immune signature mediated by oligoclonally expanded T cellsREVIEWER COMMENTS

Reviewer #1 (Remarks to the Author):

Immune-related adverse events under therapy with biologics are a heterogeneous phenomenon of high clinical relevance. Here, Bangert and colleagues describe the cellular composition in lesional skin of dupilumab-associated head and neck dermatitis (DAHND) in comparison to untreated head and neck atopic dermatitis and healthy controls. Their main finding is that DAHND is associated with a high Th22/IL-22 signature while many type 2 immune markers are downregulated. This finding is of translational relevance, as there are AD treatment options available (Jak inhibitors) or in clinical development (IL-22R antibody) that target IL-22 biology.

While relevant questions remain open - such as the question why only a few patients develop DAHND and what are early triggers - and the study has weaknesses, primarily the low number of patients in highly heterogeneous disease settings which result in lack of statistical significance in most comparisons, the study is innovative and of high interest to the broad readership of Nature Communications. The methods used are sound and the statistics valid. This study is also a rich resource for further studies and thus has a value per se.

Major comments:

Experimental revisions required:

- 1) A disadvantage of sc-RNA sequencing analysis is that many results are semi-quantitative or only relative percentages are given. In this manuscript, one example is the frequency of dendritic cells in the distinct experimental groups. The authors have the great advantage of a second cohort of patients where skin imaging can be performed. How do central findings look like in this cohort - e.g. absolute number of the main immune cell populations, absolute expression level of IL-22 or IL-13? This information would highly upvalue the manuscript.
- 2) The keratinocyte response signature to type 2 cytokines (and also type 1 and type 3 cytokines) is published (e.g. in PMID: 36513651) and could be integrated into the analysis in figure 7 to show that despite non-significant differences in IL-13 expression levels in T cells, the type 2 immune response is inhibited by dupilumab (which would be expected as the IL-4Ra should be inactivated by dupi).
- 3) Further key cytokines such as IL-17 family, IFN-g, IL-31, IL-10,... should at least be

mentioned briefly.

Interpretation:

4) Given that IL-22 is amongst the most closely correlated markers with AD severity, I think the conclusion that type 2 and type 22 immunity are "counter-regulatory" or have "antagonistic roles" is not justified by the results. It is known that Th22 cells co-express IL-13, and this finding is confirmed by the analysis here. I would agree that the results here give evidence to the fact that the trigger factors of type 2 and type 22 immunity are independent of each other.

5) It should be made clear that most comparisons did not meet statistical significance.

Kilian Eyerich

Reviewer #2 (Remarks to the Author):

The authors hope to identify mechanistic insights into the relatively common and clinically important side effect seen in some dupilumab treated AD subjects. They refer to this side effect as "dupilumab associated head and neck dermatitis" that they abbreviate as (DAHND). They studied AD subjects who developed DAHND (total n=9, n= 6 for scRNA-seq and n=3 for IF) and compared this group with scRNA-seq collected from the head/neck (n=5) or trunk (n=5) of untreated AD and healthy controls (n=4). They conclude that DAHND tissue has more IL-22 (Th22 inflammation) and reduced Type 2 inflammation.

Strengths:

1. Identifying DAHND subjects (still on Dupi) who are willing to provide a skin biopsy is difficult – so N=9 is reasonable number
2. Very little is known about what mediates this relatively common SE of dupilumab and some patients will d/c dupilumab because of it.
3. A better understanding of the genes and/or pathways in DAHND skin lesions might suggest better Rx options – especially ones that would allow AD subjects to stay on dupi
4. SC RNA-seq coupled with a small subset for which there are some IF data has the potential to yield at least pilot data on pathophysiology of this SE.

Weaknesses:

Major Concerns:

1. Most of the authors observations in DAHND came from comparing these skin samples

with the healthy control group (or AD subjects that were biopsied on a trunk lesion) and not the key comparison group which would be the AD population that underwent a head/neck skin biopsy (ideally in matching regions of the head and neck to the diseased population of interest – DAHND).

2. Lacking potentially important clinical information on AD subpopulations (DAHND, AD enrolled for the head/neck or trunk biopsy)– things like AD severity, exact anatomic area where bx was collected and how well matched these groups were with regards to some potential confounders (such as topical or systemic Rx's at the time of bx, sun/tan/photoexposure, moisturizers, bathing practices in close temporal proximity to site biopsied, etc). Without this information (and controlling for some of these that may be relevant confounders) the results are not clearly associated with DAHND (as for example they may merely reflect that DAHND skin lesions had more severe AD than the AD head and neck control group).

3. Your major conclusion is that there is an increase in IL22 – but this comes primarily from a comparison of IL22 levels in the skin of DAHND compared with skin from Healthy controls and AD Trunk biopsies (Fig 3 and 4). Therefore, you have not identified what is unique about DAHND compared with just dupilumab-untreated facial AD.

More Minor Concerns:

1. Fig 1 How do you account for the reduced “input” of cells from HC and the obvious impact that has on the % cells in cluster 1 shown in Fig 1C?

2. Fig 3 Fig 3A – How do you account for the much lower counts in HC compared to the AD subpopulations? 3H - Its concerning that some of their conclusions about the importance of AHR in DAHND development are from the T6 cluster which has only N@ 3-33 cells in AD (one of the clusters with the lowest total cell counts).

3. Fig 4 Your conclusion that clonality is more common in CD8 cells may simply reflect that you had fewer CD4 cells in the top 10 clonotypes for virtually all subjects from any of the subpopulations.

4. You made some conclusions about differences in cell types among these subpopulations but the data is pretty weak. One example of this was an increase in B cells (Fig 2) and increase in CD207+ cells (Fig 6) but, Increased B cells in the skin from DAHND could be because of higher number of cells analyzed (from each sample of DAHND vs untreated AD) (Fig 2B) and one outlier sample with high number (Fig 2H).

Reviewer #3 (Remarks to the Author):

Comment:

The manuscript of Bangert et al. analyzed the cell profiles of the lesional skin of atopic dermatitis patients with dupilumab-associated head and neck dermatitis (DAHND) and the lesions untreated utilizing single cell RNA-seq. The results showed that in DAHND, there was decreased type-2 inflammation and increased expression of Th22-related molecules, and keratinocytes expressed increased level of IL-22 receptor. The authors proposed that dupilumab inhibits the conventional type 2 inflammation in DAHND lesions, while induces hyperactivation of IL22-associated responses. The manuscript addressed an important issue in clinical practice; however, there are significant flaws in study design and data presentation of the manuscript, which hinder its publication in Nat Commun.

Special points:

- 1.The authors compared the DAHND and untreated trunk and untreated head/face, whereas the dupilumab-treated trunk was lacking. It's of vital importance to compare the difference between the face and trunk after dupilumab treatment.
- 2.Overall, the manuscript solely analyzed the sequencing data of skin lesion, but no systemically designed experimental data were provided to confirm the result.
- 3.The mechanism of increased production of Th22-related molecules in DAHND is still not clear. What's the link between anti-IL4R treatment and the observed changes in IL-22 signaling? Is there a role for IL-17 signaling?
- 4.It's also of vital importance to explain why DAHND appears mostly on the face but not other body parts of the body.
- 5.The article compared the genes of interest in each cell subpopulation (such as DC, T cells, KC, etc.) across patient groups and used violin plots to show the differences. Has the author conducted unified differential gene analysis and enrichment analysis across patient groups? In that way the authors might be able to get more hints about the differences between groups, rather than just the few selected inflammation-related genes.
- 6.Among the lymphoid populations, why were T subset-defining cytokines of the type 17 (IL17A/IL17F) or type 9 lineage (IL9) largely absent? Whereas theoretically there should be these subpopulations of T cell in human skin.
- 7.The author concluded that dupilumab effectively dampened conventional type 2

inflammation in DAHND lesions; however, in Fig. 3F-I, there was retained IL13 upregulation in multiple T-cell populations of DAHND. How to explain this phenomenon?

REVIEWER COMMENTS

Reviewer #1 (Remarks to the Author):

Immune-related adverse events under therapy with biologics are a heterogeneous phenomenon of high clinical relevance. Here, Bangert and colleagues describe the cellular composition in lesional skin of dupilumab-associated head and neck dermatitis (DAHND) in comparison to untreated head and neck atopic dermatitis and healthy controls. Their main finding is that DAHND is associated with a high Th22/IL-22 signature while many type 2 immune markers are downregulated. This finding is of translational relevance, as there are AD treatment options available (Jak inhibitors) or in clinical development (IL-22R antibody) that target IL-22 biology.

While relevant questions remain open - such as the question why only a few patients develop DAHND and what are early triggers - and the study has weaknesses, primarily the low number of patients in highly heterogeneous disease settings which result in lack of statistical significance in most comparisons, the study is innovative and of high interest to the broad readership of Nature Communications. The methods used are sound and the statistics valid. This study is also a rich resource for further studies and thus has a value per se.

Response: We thank the reviewer for these encouraging comments.

Major comments:

Experimental revisions required:

1) A disadvantage of sc-RNA sequencing analysis is that many results are semi-quantitative or only relative percentages are given. In this manuscript, one example is the frequency of dendritic cells in the distinct experimental groups. The authors have the great advantage of a second cohort of patients where skin imaging can be performed. How do central findings look like in this cohort - e.g. absolute number of the main immune cell populations, absolute expression level of IL-22 or IL-13? This information would highly upvalue the manuscript.

Response: We thank the reviewer for these important remarks. In addition to relative numbers in Figure 1C, we have now added absolute numbers of myeloid cell populations in Figure 5C. We agree that due to the small sample size, key findings from single-cell RNA sequencing analyses need to be corroborated in independent sample sets. For this reason, we have initially quantified numbers of key cells including B cells (Figure 2H), Langerhans cells (Figure 6) and cytotoxic T cells (Figure 4) by multicolor immunofluorescence in these independent sample sets, and numbers were largely in line with scRNAseq data. In addition, we have now added quantitative RT-PCR analyses of bulk mRNA isolates from such independent biopsies, comprising healthy control skin, untreated trunk AD, dupilumab-treated trunk AD, untreated head-neck AD, as well as DAHND. From these samples, we have quantified mRNA levels of the most prominent T-

cell cytokines (*IFNG*, *IL13*, *IL22*, *IL26*, *IL4*, *IL5*, *IL31*, *IL9*, *IL17A*, *IL17F*, *IL10*) and now display these data in a newly added Supplementary Figure S1. Due to the difficulty of finding patients willing to donate head and neck skin biopsies, sample numbers are still low, but nevertheless, quantitative RT-PCR results are highly consistent with single-cell RNA sequencing data. Most importantly, we observe sustained upregulation of *IL13* in DAHND lesions, comparable to the levels of untreated AD skin, while we could confirm further upregulation of *IL22* in DAHND lesions. Given this independent validation cohort, and the consistent immunofluorescence and quantitative RT-PCR data, we are confident cell numbers in our scRNAseq results are indeed valid.

2) The keratinocyte response signature to type 2 cytokines (and also type 1 and type 3 cytokines) is published (e.g. in PMID: 36513651) and could be integrated into the analysis in figure 7 to show that despite non-significant differences in IL-13 expression levels in T cells, the type 2 immune response is inhibited by dupilumab (which would be expected as the IL-4Ra should be inactivated by dupi).

Response: We thank the reviewer for pointing out this study, that comprehensively combines spatial transcriptomics with *in vitro* keratinocyte stimulation results. We have now integrated the type 2-specific keratinocyte responder gene signature into our Figure 7E and can indeed see that these genes show a trend of downregulation in keratinocytes in DAHND as opposed to untreated AD samples.

3) Further key cytokines such as IL-17 family, IFN-g, IL-31, IL-10,... should at least be mentioned briefly.

Response: Aside from the brief scRNA-seq description of these cytokines in Figure 3C, we have now added quantitative RT-PCR data in a new Supplementary Figure S1 showing multiple T cell cytokines within various disease groups, in order to corroborate scRNA-seq data.

Interpretation:

4) Given that IL-22 is amongst the most closely correlated markers with AD severity, I think the conclusion that type 2 and type 22 immunity are "counter-regulatory" or have "antagonistic roles" is not justified by the results. It is known that Th22 cells co-express IL-13, and this finding is confirmed by the analysis here. I would agree that the results here give evidence to the fact that the trigger factors of type 2 and type 22 immunity are independent of each other.

Response: We have now removed this comment from the abstract, and modified it in the discussion section and the "One sentence summary".

5) It should be made clear that most comparisons did not meet statistical significance.

Response: We agree that due to small sample size, many comparisons did not yield statistical significance, but all key findings, including IL13 and IL22 regulation in T cells, were substantiated by the presence of these genes within the top differentially expressed genes between DAHND and other groups, as listed in Supplementary Table S3.

Reviewer #2 (Remarks to the Author):

The authors hope to identify mechanistic insights into the relatively common and clinically important side effect seen in some dupilumab treated AD subjects. They refer to this side effect as “dupilumab associated head and neck dermatitis” that they abbreviate as (DAHND). They studied AD subjects who developed DAHND (total n=9, n= 6 for scRNA-seq and n=3 for IF) and compared this group with scRNA-seq collected from the head/neck (n=5) or trunk (n=5) of untreated AD and healthy controls (n=4). They conclude that DAHND tissue has more IL-22 (Th22 inflammation) and reduced Type 2 inflammation.

Strengths:

1. Identifying DAHND subjects (still on Dupi) who are willing to provide a skin biopsy is difficult – so N=9 is reasonable number
2. Very little is known about what mediates this relatively common SE of dupilumab and some patients will d/c dupilumab because of it.
3. A better understanding of the genes and/or pathways in DAHND skin lesions might suggest better Rx options – especially ones that would allow AD subjects to stay on dupi
4. SC RNA-seq coupled with a small subset for which there are some IF data has the potential to yield at least pilot data on pathophysiology of this SE.

Response: We thank the reviewer for these encouraging comments.

Weaknesses:

Major Concerns:

1. Most of the authors observations in DAHND came from comparing these skin samples with the healthy control group (or AD subjects that were biopsied on a trunk lesion) and not the key comparison group which would be the AD population that underwent a head/neck skin biopsy (ideally in matching regions of the head and neck to the diseased population of interest – DAHND).

Response: We completely agree with the reviewer’s comment that the comparison between DAHND and head-neck AD lesions is key for this project. In our analyses, we in fact included a comparison of untreated AD from the head-neck region with location-matched DAHND lesions (initial Table S6). In addition, and to reduce complexity of multiple pairwise comparisons, we decided to compare all 4 groups (HC, trunk AD, head neck AD as well as DAHND) within a single Supplementary Table S3 to demonstrate

unique regulation for each group in comparison to all other groups taken together. We have now moved data from the initial Table S6 to a new Table S4 to place it directly after Table S3, to make it more convenient for the reader to directly compare. Importantly, this Table S4 essentially corroborates findings in Table S3. We are thus confident that our approach in Table S3 is indeed comprehensively assessing significant gene regulation.

2. Lacking potentially important clinical information on AD subpopulations (DAHND, AD enrolled for the head/neck or trunk biopsy)– things like AD severity, exact anatomic area where bx was collected and how well matched these groups were with regards to some potential confounders (such as topical or systemic Rx at the time of bx, sun/tan/photoexposure, moisturizers, bathing practices in close temporal proximity to site biopsied, etc). Without this information (and controlling for some of these that may be relevant confounders) the results are not clearly associated with DAHND (as for example they may merely reflect that DAHND skin lesions had more severe AD than the AD head and neck control group).

Response: We are thankful for this important remark. When obtaining skin biopsies, patients did not use any concurrent systemic medication or experienced photo-exposure as indicated in Table 1. Most patients used moisturizers on a regular basis as part of their treatment schedule. A few patients did use topical treatment intermittently, but not within 4 days or more before we obtained the biopsy sample. We have now expanded Table 1 with additional pertinent information and also included IGA as an objective score for AD severity.

3. Your major conclusion is that there is an increase in IL22 – but this comes primarily from a comparison of IL22 levels in the skin of DAHND compared with skin from Healthy controls and AD Trunk biopsies (Fig 3 and 4). Therefore, you have not identified what is unique about DAHND compared with just dupilumab-untreated facial AD.

Response: Thank you for this observation, we would like to clarify our findings. We also analyzed head-neck skin from untreated AD patients (“head/neck AD” group across all figures). DAHND lesions showed significant IL22 upregulation in DEG of T cell clusters T1 and T4 as evidenced by significant regulation in Supplementary Tables S3 and S4. In addition, we have substantiated these claims in independent sample sets by performing multicolor immunofluorescence (Figures 4E-F and J-K), as well as quantitative RT-PCR in a newly added Supplementary Figure S1, that all compare untreated head-neck AD with DAHND lesions as part of the analysis.

More Minor Concerns:

1. Fig 1 How do you account for the reduced “input” of cells from HC and the obvious impact that has on the % cells in cluster 1 shown in Fig 1C?

Response: Reduced numbers of cells in HC samples can indeed be a challenge, and we optimized our isolation protocols in several previous studies (Rojahn et al, PMID

32344053, Rindler et al., PMID 33717163) to make sure that our scRNAseq datasets indeed reflect true in situ cellularity. Nevertheless, given that normal skin is less inflamed, there are of course still lower cell numbers in multiple cell groups. Statistically, we identified differentially expressed genes (DEGs) between two groups of cells by using a Wilcoxon Rank Sum test, also known as 'Mann-Whitney' test, which, as a non-parametric test, is comparably robust with different sample sizes, although statistical power diminishes as the group sizes become more unequal. We thus assessed our critical scRNAseq findings in independent sample sets using multicolor immunofluorescence stainings (Figure 2G-H, Figure 4 E-F and J-K, Figure 6) and quantitative RT-PCR (Figure S1), which corroborated all scRNA-seq results.

2. Fig 3: Fig 3A – How do you account for the much lower counts in HC compared to the AD subpopulations?

Response: We assessed our critical scRNAseq findings in independent sample sets using multicolor immunofluorescence stainings (Figure 2G-H, Figure 4 E-F and J-K, Figure 6) and quantitative RT-PCR (Figure S1), which corroborated all scRNA-seq results.

3H - Its concerning that some of their conclusions about the importance of AHR in DAHND development are from the T6 cluster which has only N@ 3-33 cells in AD (one of the clusters with the lowest total cell counts).

Response: We agree with the reviewer that T6 is one of the smallest T cell clusters. While it is challenging to determine a specific universal threshold, a cluster size above 100 cells is generally considered sufficient to characterize a phenotype in droplet-based scRNA-seq analyses. In total, our cluster T6 contains 224 cells and, as described above, we applied a Wilcoxon Rank Sum test with Bonferroni correction to find differentially expressed genes. Moreover, in addition to upregulation of *AHR* in this small cluster, we also found *AHR* to be among the top DEG in larger clusters such as T1 and T4 (see Supplementary Table S3). Therefore, we are confident that our findings indeed reflect pathophysiological processes in the general T cell population of DAHND lesions.

3. Fig 4 Your conclusion that clonality is more common in CD8 cells may simply reflect that you had fewer CD4 cells in the top 10 clonotypes for virtually all subjects from any of the subpopulations.

Response: We believe due to its high consistency that this distribution is not random, but that indeed there are more CD8 cells in the top clonotypes in particular disease subsets, while there is a more balanced CD4 / CD8 ratio of the top 10 clones in healthy control individuals.

4. You made some conclusions about differences in cell types among these subpopulations but the data is pretty weak. One example of this was an increase in B cells (Fig 2) and increase in CD207+ cells (Fig 6) but, Increased B cells in the skin from DAHND could be because of higher number of cells analyzed (from each

sample of DAHND vs untreated AD) (Fig 2B) and one outlier sample with high number (Fig 2H).

Response: We agree that single-cell counts are prone to bias from isolation techniques. For this reason, we have confirmed cell counts for key cell subtypes in independent samples using multicolor immunofluorescence, which corroborated our single-cell data. From these counts we extrapolate that overall cell numbers obtained in single-cell RNA sequencing analyses are indeed reflective of true cell counts in the lesions.

Reviewer #3 (Remarks to the Author):

Comment:

The manuscript of Bangert et al. analyzed the cell profiles of the lesional skin of atopic dermatitis patients with dupilumab-associated head and neck dermatitis (DAHND) and the lesions untreated utilizing single cell RNA-seq. The results showed that in DAHND, there was decreased type-2 inflammation and increased expression of Th22-related molecules, and keratinocytes expressed increased level of IL-22 receptor. The authors proposed that dupilumab inhibits the conventional type 2 inflammation in DAHND lesions, while induces hyperactivation of IL22-associated responses. The manuscript addressed an important issue in clinical practice; however, there are significant flaws in study design and data presentation of the manuscript, which hinder its publication in Nat Commun.

Special points:

1. The authors compared the DAHND and untreated trunk and untreated head/face, whereas the dupilumab-treated trunk was lacking. It's of vital importance to compare the difference between the face and trunk after dupilumab treatment.

Response: We thank the reviewer for this important suggestion. We have now added five samples from trunk skin of AD patients successfully treated with dupilumab, and added these data to a new Supplementary Figure S1 showing RT-PCR data. In these samples we find that proinflammatory cytokines, including IL13 and IL22, are considerably decreased in successfully treated trunk AD when compared to untreated samples as well as DAHND lesions.

2. Overall, the manuscript solely analyzed the sequencing data of skin lesion, but no systemically designed experimental data were provided to confirm the result.

Response: Our aim was to profile DAHND within patients, to better understand the molecular landscape of these lesions. We believe that our manuscript indeed contains functional systemic data, as we compare patients with or without exposure to a targeted immunomodulatory agent (i.e. dupilumab). We are not aware of any animal model system that can sufficiently reflect the appearance of DAHND in all its complexity. This is why we entirely rely on the actual human disease in this project.

3.The mechanism of increased production of Th22-related molecules in DAHND is still not clear. What's the link between anti-IL4R treatment and the observed changes in IL-22 signaling? Is there a role for IL-17 signaling?

Response: We believe that in some individuals, the blockade of IL4R α might lead to oligoclonal and, thus likely, antigen-specific expansion of IL22-producing AHR+ T cells as a result of exaggerated antimicrobial defense mechanisms to otherwise harmless commensals in the facial area such as *Malassezia spp* or *Demodex* mites. Some findings from the literature can be interpreted as supporting this hypothesis. It has been shown that the *Malassezia sympoidalis*-derived allergen Mala s13 has great homology to stress-induced human thioredoxin and thereby may function as an autoreactive T-cell antigen (Balaji et al.: PMID 21489611). Skin-derived Mala S13-reactive T-cells contribute to skin inflammation by producing cytokines such as IFN- γ , IL-13, IL-17 or IL-22. The absence of type 2 immunity can also lead to *Demodex* outgrowth as recently demonstrated in IL4ra^{-/-} mice (Ricardo-Gonzalez et al.: PMID 36044899). Consistently, patients with severe *Demodex*-associated skin diseases such as rhinophyma show increased expression levels of IL-22 and IFN- γ as well as decreased IL-13 levels within the skin (Ricardo-Gonzalez et al.: PMID 36044899).

We can currently only speculate that DAHND reflects an IL-22 dominated hypersensitivity reaction to commensal colonization in the absence of functioning IL-13 signaling. Future experiments aiming at the identification of *Malassezia*-specific T-cell clones from DAHND lesions are needed to provide further pathomechanistic insights.

In addition to its pathogenic role in atopic diseases, type 2 inflammation may have a regulatory effect on other T cell arms including type 22 cells. Importantly, we did not find significantly increased levels of IL17A or IL17F in our samples, a finding that we now corroborate in additional quantitative RT-PCR datasets in a newly added Supplementary Figure S1. Thus, we do not believe that IL-17 production has a significant role in this process. We added a paragraph further addressing this issue in the discussion section (page 20-21).

4.It's also of vital importance to explain why DAHND appears mostly on the face, but not other body parts of the body.

Response: We agree that this is a key question, but this question remains currently unexplained. Based on our current knowledge we propose that an IL-22 driven hypersensitivity reaction to commensals such as *Malassezia spp* or *Demodex* mites might be the cause for DAHND, as suggested by other authors (Kozera et al.: PMID 35588929, Muzumdar et al.: PMID 34855151), which we now also discuss in the manuscript (page 21). As *Malassezia spp* and *Demodex* are predominantly found in the human head-neck area, this observation would explain the regional occurrence of DAHND in affected patients.

5. The article compared the genes of interest in each cell subpopulation (such as DC, T cells, KC, etc.) across patient groups and used violin plots to show the differences. Has the author conducted unified differential gene analysis and

enrichment analysis across patient groups? In that way the authors might be able to get more hints about the differences between groups, rather than just the few selected inflammation-related genes.

Response: We have now added such an analysis in Supplementary Table S7. These data show that among the 3 top upregulated genes in DAHND we find *IL22*, as well as the IL-22 dependent inflammatory markers *S100A7* and *S100A9*. Therefore, we are confident that our hypothesis of IL-22-driven immunity in DAHND is in fact substantiated by our data. We also tried to perform pathway enrichment analyses. However, these did not yield meaningful results, as they are more designed for distinct diseases rather than more nuanced differences within related diseases.

6. Among the lymphoid populations, why were T subset-defining cytokines of the type 17 (IL17A/IL17F) or type 9 lineage (IL9) largely absent? Whereas theoretically there should be these subpopulations of T cell in human skin.

Response: We agree that previously, IL22 production was linked to type 17 immune activation (Liang et al.: PMID 16982811), but more recent single-cell analyses showed that IL22 can be largely co-produced by type 2 cells, independent of IL17 production (e.g. Bangert et al.: PMID 33483337). However, given the fact that single-cell RNA sequencing datasets can have limited sensitivity, we have now included quantitative RT-PCR data from an independent sample set in a new Supplementary Figure S1. Importantly, these data corroborate our single-cell RNA sequencing findings of a lack or low expression levels of IL17A or IL17F production in DAHND.

7. The author concluded that dupilumab effectively dampened conventional type 2 inflammation in DAHND lesions; however, in Fig. 3F-I, there was retained IL13 upregulation in multiple T-cell populations of DAHND. How to explain this phenomenon?

Response: These data are consistent to what we and others have shown before (Bangert et al.: PMID:33483337; Ariens et al.: PMID 31593343). Even upon one year of successful dupilumab treatment and clearance of skin disease, T cells maintain some IL-13 production, but this IL-13 does not elicit downstream inflammatory responses due to blockade of its receptor, as evidenced by a decrease in type 2-associated chemokines including CCL17. This persistent IL-13 upregulation might constitute the AD disease memory, because upon cessation of any treatment, AD typically recurs sooner or later.

REVIEWERS' COMMENTS

Reviewer #1 (Remarks to the Author):

The authors have added substantial new data (bulk sequencing data, data from lesional AD skin under dupilumab at the trunk, imaging data) and therefore clearly improved the manuscript regarding validity. The major finding of increased IL-22 in DAHND lesions is confirmed and a potential mechanistic basis, namely *Malassezia* overgrowth, is reasonably hypothesized. All conclusions are justified from the data. I congratulate the authors to this important and clinically relevant study and recommend publication as is.

Reviewer #2 (Remarks to the Author):

The authors have done a very nice job responding to the 3 reviewers and the topic is of great interest and import with the widespread use of dupilumab in AD pts and the relatively frequent appearance of a not fully understood head and neck dermatitis (DAHND). The magnitude of analysis performed is noteworthy.

The question that remains is whether there are sufficient number of samples from DAHND and the only really relevant and realistic comparison group - of AD pts with head and neck dermatitis but without dupilumab exposure- to offer firm conclusions.

Reviewer #3 (Remarks to the Author):

All the concerns have been addressed, and I have no further questions.